# SUBZERO: RANDOM SUBSPACE ZEROTH-ORDER OPTIMIZATION FOR MEMORY-EFFICIENT LLM FINE-TUNING

## ABSTRACT

Fine-tuning Large Language Models (LLMs) has proven effective for a variety of downstream tasks. However, as LLMs grow in size, the memory demands for backpropagation become increasingly prohibitive. Zeroth-order (ZO) optimization methods offer a memory-efficient alternative by using forward passes to estimate gradients, but the variance of gradient estimates typically scales linearly with the model's parameter dimension—a significant issue for LLMs. In this paper, we propose the random Subspace Zeroth-order (SubZero) optimization to address the challenges posed by LLMs' high dimensionality. We introduce a low-rank perturbation tailored for LLMs that significantly reduces memory consumption while improving training performance. Additionally, we prove that our gradient estimation closely approximates the backpropagation gradient, exhibits lower variance than traditional ZO methods, and ensures convergence when combined with SGD. Experimental results show that SubZero enhances fine-tuning performance and achieves faster convergence compared to standard ZO approaches like MeZO across various language modeling tasks. The source code will be released publicly.

## 1 INTRODUCTION

Large Language Models (LLMs), such as the GPT and LLaMA series (Zhang et al., 2022; Touvron et al., 2023), have recently demonstrated impressive capabilities in natural language processing tasks and beyond (Solaiman et al., 2019; Achiam et al., 2023). These models utilize deep learning, particularly the transformer architecture (Vaswani et al., 2017), to learn complex patterns in language data. However, LLMs can struggle with specialized tasks that require domain-specific knowledge (Shen et al., 2024). Fine-tuning presents an effective solution by slightly adjusting pre-trained LLMs with domain data, enabling them to adapt to specific tasks more effectively.

For fine-tuning, first-order (FO) optimizers, such as SGD (Amari, 1993) or Adam (Kingma & Ba, 2015), are commonly used to achieve promising performance on domain datasets. However, as LLMs grow in size, FO optimizers demand increasingly memory consumption due to the gradient computations required by backpropagation (BP) (Zhao et al., 2024a). To enhance memory efficiency, MeZO (Malladi et al., 2023) first introduces the zeroth-order (ZO) optimizer to LLM fine-tuning without BP. It just needs forward passes and calculates gradient estimates using finite differences of training loss values. Nevertheless, the variance of ZO gradient estimates linearly depends on the perturbation dimension, which corresponds to the number of model parameters. This can become extremely large in LLMs, resulting in significant performance degradation compared to FO optimizers (Gautam et al., 2024; Jiang et al., 2024; Liu et al., 2024).

There are two main attempts to addressing the high variance of ZO gradient estimates. The first approach involves increasing batch size alongside training iterations, which reduces gradient noise and variance in ZO gradient estimates (Gautam et al., 2024; Jiang et al., 2024). However, this leads to significant runtime and memory costs due to the large batch size in the later training stages. The second approach focuses on perturbing fewer parameters by employing sparse parameter perturbations, such as random and sparse pruning masks (Liu et al., 2024) and block-coordinate perturbations (Zhang et al., 2024), or by reducing the number of trainable parameters through techniques like parameter-efficient fine-tuning (PEFT) (Malladi et al., 2023; Zhang et al., 2024) and tensorized adapters (Yang

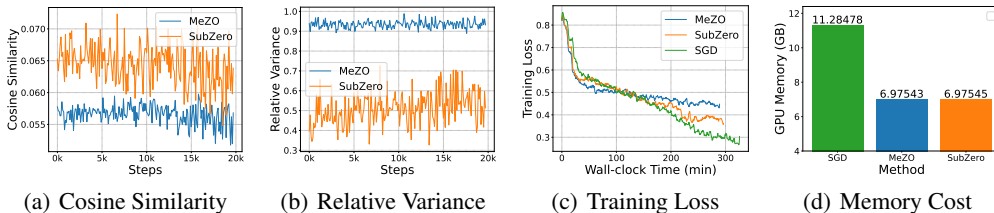

(a) Cosine Similarity  (b) Relative Variance  (c) Training Loss  (d) Memory Cost

Figure 1: Visualization of cosine similarity $\mathbb{E}\left[\mathtt{cosine}(\boldsymbol{g}, \hat{\boldsymbol{g}})\right]$, relative variance $\mathrm{Var}\left[\|\hat{\boldsymbol{g}}\|\right]/\|\boldsymbol{g}\|^2$, training loss, and GPU memory cost on OPT-1.3B under the prompt tuning scheme. Here, $\hat{\boldsymbol{g}}$ represents the gradient estimated by MeZO or our SubZero, and $\boldsymbol{g}$ denotes the expected gradient $\mathbb{E}[\hat{\boldsymbol{g}}]$. SubZero demonstrates reduced angle error and variance in gradient estimation, while also accelerating convergence with minimal additional memory overhead.

et al., 2024). Recent theoretical advancements have proposed using random projections to lessen the dimensionality dependence in ZO optimizers (Nozawa et al., 2024; Roberts & Royer, 2023; Kozak et al., 2021) by applying low-dimensional perturbations in random subspaces. Nonetheless, a major drawback of this approach is the need to store a huge projection matrix that scales with model parameter dimensionality, making it impractical for fine-tuning large LLMs.

**Contributions.** In this work, we propose the first random Subspace Zeroth-order (SubZero) optimization to tackle the challenges of high-dimensional LLM fine-tuning. We introduce a low-rank perturbation to estimate the gradient, specifically designed for LLM architecture, leading to reduced memory consumption and enhanced training performance. Our main contributions are as follows.

Firstly, we propose a layer-wise low-rank perturbation approach for gradient estimation, specifically designed for fine-tuning LLMs. In each layer, we generate a low-rank perturbation matrix by combining two column-orthogonal matrices with a Gaussian random matrix, which is then used for gradient estimation. Unlike traditional ZO methods like MeZO (Malladi et al., 2023) which apply non-low-rank perturbations to the entire model, our approach significantly reduces the variance of gradient estimates and the angle error between the estimated gradient and its expectation, as respectively shown in Fig. 1 (a) and (b). SubZero also improves upon random subspace ZO methods like S-RGF (Nozawa et al., 2024) by using smaller and layer-specific low-rank perturbation matrices instead of a large and model-scale projection matrix, thus cutting memory and computational costs. Additionally, we introduce a lazy update strategy, generating perturbations periodically rather than iteratively, further reducing overhead. Besides, we also successfully apply SubZero to four popular LLM fine-tuning schemes, highlighting the compatibility of SubZero.

Secondly, we provide theoretical guarantees for SubZero. We first convert our gradient estimation into an equivalent formulation, highlighting the key differences between our approach and existing traditional ZO methods (Malladi et al., 2023), as well as random subspace ZO methods (Nozawa et al., 2024). Then, we prove that the gradient estimated by SubZero closely approximates the BP gradient, i.e., the ground-truth gradient, and enjoys significantly lower gradient variance than traditional ZO methods like MeZO. Furthermore, we establish the theoretical convergence of SubZero when combined with the SGD optimizer.

Finally, experimental results demonstrate SubZero's superior performance and memory efficiency compared to other ZO approaches in both full-parameter tuning and parameter-efficient fine-tuning (PEFT) schemes, such as LoRA, prefix tuning, and prompt tuning. For instance, SubZero improves upon MeZO by 7.1% on LLaMA-7B and by 3.2% on OPT-1.3B under full-parameter tuning and prompt tuning, while maintaining nearly identical memory costs to MeZO.

## 2 RELATED WORK

**Zeroth-Order Fine-Tuning.** ZO optimizers utilize just two forward passes to estimate gradient without BP. Malladi et al. (2023) first used ZO optimization to fine-tune LLMs, significantly lowering the GPU hours and memory usage to levels similar to inference, which offers a considerable advantage over FO optimizers. They demonstrated that LLM fine-tuning benefits from a well-structured loss

landscape by introducing suitable task-specific prompt templates. Convergence theories for ZO optimization have been elaborated in both convex (Nesterov & Spokoiny, 2017; Jamieson et al., 2012; Duchi et al., 2015) and non-convex settings (Liu et al., 2018; Ji et al., 2019). However, these convergence rates typically increase linearly with the number of trainable parameters (Nesterov & Spokoiny, 2017; Jamieson et al., 2012; Duchi et al., 2015; Liu et al., 2018; Ji et al., 2019).

Recently, more work in ZO has focused on improving the convergence rates and reducing gradient estimation variance for LLM fine-tuning. Increasing batch size can diminish noise in ZO gradient estimation (Gautam et al., 2024; Jiang et al., 2024). Perturbing a subset of model parameters also lowers gradient variance. This approach induces sparse parameter perturbations through random and sparse pruning masks (Liu et al., 2024) or block-coordinate perturbations (Zhang et al., 2024). Additionally, some approaches tried to reduce trainable parameters through PEFT (Malladi et al., 2023; Zhang et al., 2024) and tensorized adapters (Yang et al., 2024).

**Random Subspace Optimization.** To lessen dependence on dimensionality, some research utilizes random projections and low-dimensional perturbations in subspaces (Nozawa et al., 2024; Roberts & Royer, 2023; Kozak et al., 2021). However, these methods are hindered by the need to store a large projection matrix that increases with dimensionality, making it impractical for fine-tuning LLMs.

**Memory-Efficient Fine-Tuning.** Fine-tuning generally employs FO optimizers like SGD (Amari, 1993) or Adam (Kingma & Ba, 2015). Various approaches have been developed to reduce the memory cost of BP, such as sparsifying gradients (Sun et al., 2017), projecting gradients into a low-rank subspace (Zhao et al., 2024a), and quantizing optimizer states to lower bits (Dettmers et al., 2022b; Li et al., 2024). Additional methods to conserve activation and weight memory during forward and backward passes include gradient checkpointing (Chen et al., 2016), FlashAttention (Dao et al., 2022), QLoRA (Dettmers et al., 2024), and LLM.int8() (Dettmers et al., 2022a).

## 3 PRELIMINARIES

In this section, we introduce the most popular ZO optimization approach and existing random subspace optimization methods.

**Notations.** We use a non-bold letter like $a$ and $A$ to denote a scalar, a boldfaced lower-case letter like $\boldsymbol{w}$ to denote a column vector, and a boldfaced upper-case letter such as $\boldsymbol{W}$ to denote a matrix. $\mathcal{N}(\boldsymbol{0}, \boldsymbol{I})$ denotes a multivariate normal distribution with a zero mean vector and an identity covariance matrix. $\text{vec}(\boldsymbol{W})$ represents the vectorization of matrix $\boldsymbol{W}$, which transforms $\boldsymbol{W}$ into a column vector by stacking the columns of $\boldsymbol{W}$ vertically. $\boldsymbol{A} \otimes \boldsymbol{B}$ is the Kronecker product of matrices $\boldsymbol{A}$ and $\boldsymbol{B}$. $\mathbb{E}[\boldsymbol{x}]$ represents the expected value of a random variable $\boldsymbol{x}$. $\text{Var}[\boldsymbol{x}]$ represents the variance of a random variable $\boldsymbol{x}$. The $\ell_2$-norm of a vector $\boldsymbol{x}$ is $\|\boldsymbol{x}\| = \sqrt{\sum_{i=1}^{n} \boldsymbol{x}_i^2}$. The spectral norm of a matrix $\boldsymbol{A}$ is $\|\boldsymbol{A}\|$. The Frobenius norm of a matrix $\boldsymbol{A}$ is $\|\boldsymbol{A}\|_F = \sqrt{\langle \boldsymbol{A}, \boldsymbol{A} \rangle}$. $C_L^{s,p}(\mathcal{S})$ represents the class of $s$-th smooth and $p$-th $L$-smooth functions over the set $\mathcal{S}$. $\text{bdiag}(\boldsymbol{A}_1, \boldsymbol{A}_2, \cdots, \boldsymbol{A}_l)$ is a block diagonal matrix with diagonal blocks $\boldsymbol{A}_1, \boldsymbol{A}_2, \cdots, \boldsymbol{A}_l$.

We are interested in fine-tuning large LLMs (Ding et al., 2023). These models typically comprise multiple layers, with trainable parameter vectors represented as $\boldsymbol{w} = \left[\boldsymbol{w}_1^\mathsf{T}, \boldsymbol{w}_2^\mathsf{T}, \ldots, \boldsymbol{w}_l^\mathsf{T}\right]^\mathsf{T} \in \mathbb{R}^d$, where $\boldsymbol{w}_i$ denotes the flattened parameter vector from the $i$-th layer and $d$ is the number of model parameters. Then training these models involves optimizing the following problem:

$$\min_{\boldsymbol{w}} \mathcal{L}(\boldsymbol{w}), \tag{1}$$

where $\mathcal{L}(\cdot)$ denotes the loss function.

**Zeroth-Order Optimization.** ZO optimization is BP-free and estimates gradients via random perturbations. A classical gradient estimator is the simultaneous perturbation stochastic approximation (SPSA) (Spall, 1992), which is defined as

$$\widehat{\nabla}\mathcal{L}(\boldsymbol{w}; \mathcal{B}) = \frac{\mathcal{L}(\boldsymbol{w} + \varepsilon\boldsymbol{z}; \mathcal{B}) - \mathcal{L}(\boldsymbol{w} - \varepsilon\boldsymbol{z}; \mathcal{B})}{2\varepsilon}\boldsymbol{z}, \tag{2}$$

where $\mathcal{L}(\boldsymbol{w}; \mathcal{B})$ is the loss on a minibatch $\mathcal{B}$ of size $B$ uniformly sampled from the training dataset $\mathcal{D}$, $\boldsymbol{z} \in \mathbb{R}^d$ represents a random perturbation sampled from $\mathcal{N}(\boldsymbol{0}, \boldsymbol{I}_d)$, and $\varepsilon$ is the perturbation scale.

The SPSA in Eqn. (2) is an unbiased gradient estimator of the desired gradient $\nabla\mathbb{E}_{\boldsymbol{z}}[\mathcal{L}(\boldsymbol{w} + \varepsilon\boldsymbol{z})]$ (Nesterov & Spokoiny, 2017). It only requires two forward passes to estimate the gradient and eliminates

the need for BP computation, resulting in substantial savings in computation cost and GPU memory usage. With this estimated gradient, it is easy to integrate with existing FO optimizers like SGD and develop corresponding ZO optimizers, such as ZO-SGD defined as:

$$\boldsymbol{w}^{t+1} = \boldsymbol{w}^t - \eta^t \widehat{\nabla}\mathcal{L}(\boldsymbol{w}^t; \mathcal{B}^t), \tag{3}$$

where $\eta^t > 0$ is the learning rate at iteration $t$. To boost memory efficiency, MeZO (Malladi et al., 2023) implements ZO-SGD via in-place operations and employs a single random seed to facilitate efficient perturbation regeneration, significantly reducing memory overhead.

**Random Subspace Optimization.** Recent theoretical work (Nozawa et al., 2024; Roberts & Royer, 2023) has explored using low-dimensional perturbations in random subspaces to reduce gradient variances and enhance convergence rates. The key to random subspace methods is the generation of the perturbation vector $\tilde{z}$ within a subspace spanned by $\boldsymbol{P}$:

$$\tilde{z} = \boldsymbol{P}z, \tag{4}$$

where $\boldsymbol{P} \in \mathbb{R}^{d \times q}$ is a random projection matrix with entries drawn from $\mathcal{N}(0, 1)$, $\boldsymbol{z} \in \mathbb{R}^q$ is a low-dimensional random perturbation vector sampled from $\mathcal{N}(\boldsymbol{0}, \boldsymbol{I}_q)$, and $q < d$ is the dimension of the subspace. Thus, the gradient estimator in the subspace is given as follows:

$$\widehat{\nabla}\mathcal{L}(\boldsymbol{w}, \boldsymbol{P}; \mathcal{B}) = \frac{\mathcal{L}(\boldsymbol{w} + \varepsilon\boldsymbol{P}z; \mathcal{B}) - \mathcal{L}(\boldsymbol{w} - \varepsilon\boldsymbol{P}z; \mathcal{B})}{2\varepsilon}\boldsymbol{P}z. \tag{5}$$

LLMs have a large model size, and thus their training and fine-tuning parameters can be very high-dimensional. This results in an excessively large matrix $\boldsymbol{P}$ which is $q$ times larger than the model size $d$ in full-parameter tuning (Aghajanyan et al., 2021) and is also large in other fine-tuning schemes e.g., LoRA (Hu et al., 2022). Consequently, this approach significantly increases memory requirements and computational complexity. Therefore, it is crucial to develop an efficient subspace construction strategy with minimal memory consumption for LLM fine-tuning.

## 4 METHODOLOGY

Here we first elaborate on our SubZero, a powerful ZO framework designed for LLM fine-tuning. Then we present how to integrate SubZero into four representative fine-tuning schemes.

### 4.1 RANDOM SUBSPACE OPTIMIZATION FOR LLM FINE-TUNING

Our intuition is that exploring update directions in a low-dimensional subspace may result in a reduced variance of the estimated gradient compared to the estimation in the vanilla space as used in MeZO. Inspired by (Zhao et al., 2024a; Nozawa et al., 2024; Roberts & Royer, 2023), we propose the random Subspace Zeroth-order (SubZero) optimization framework tailored for LLM fine-tuning. This framework reduces gradient estimation variance, and minimizes the memory overhead associated with gradient estimation, such as the memory overhead caused by the projection matrix $\boldsymbol{P}$ in Eqn. (5) used in (Nozawa et al., 2024; Roberts & Royer, 2023).

**Layer-wise Random Subspace Perturbation.** LLMs primarily consist of dense layers that perform matrix multiplication. We denote the trainable parameters of the $i$-th layer in matrix form as $\boldsymbol{W}_i \in \mathbb{R}^{m_i \times n_i}$. Then we will explain how to design its low-rank perturbation $\tilde{\boldsymbol{Z}}_i \in \mathbb{R}^{m_i \times n_i}$.

We propose a low-rank perturbation strategy for model parameter matrix of each layer, contrasting with previous random subspace methods that focus on the entire model's parameters (Nozawa et al., 2024; Roberts & Royer, 2023). At each iteration, we generate a low-dimensional random matrix $\boldsymbol{Z}_i \in \mathbb{R}^{r \times r}$, where $r \ll \min\{m_i, n_i\}$, and perform QR decomposition on two random matrices to create projection matrices $\boldsymbol{U}_i \in \mathbb{R}^{m_i \times r}$ and $\boldsymbol{V}_i \in \mathbb{R}^{n_i \times r}$ (see Algorithm 1). Both $\boldsymbol{U}_i$ and $\boldsymbol{V}_i$ are column-orthogonal matrices. Our experiments in Table 6 indicate that using Gaussian random projection matrices yields worse performance than using our designed column-orthogonal matrices. Then we combine these three matrices to yield a low-rank perturbation as follows:

$$\tilde{\boldsymbol{Z}}_i = \boldsymbol{U}_i\boldsymbol{Z}_i\boldsymbol{V}_i^\mathsf{T}, \tag{6}$$

where $\tilde{\boldsymbol{Z}}_i$ is the perturbation matrix in a subspace spanned by $\boldsymbol{U}_i$ and $\boldsymbol{V}_i$, and $\boldsymbol{Z}_i$ represents the low-dimensional random perturbation matrix with entries sampled from $\mathcal{N}(0,1)$.

Let the model consist of $l$ layers, with the parameter matrix set defined as $\mathcal{W} = \{\boldsymbol{W}_i\}_{i=1}^l$ and the perturbation matrix set as $\tilde{\mathcal{Z}} = \{\tilde{\boldsymbol{Z}}_i\}_{i=1}^l$. Similar to Eqns. (2) and (5), we compute the loss difference:

$$\rho = \frac{\mathcal{L}(\mathcal{W} + \varepsilon\tilde{\mathcal{Z}}; \mathcal{B}) - \mathcal{L}(\mathcal{W} - \varepsilon\tilde{\mathcal{Z}}; \mathcal{B})}{2\varepsilon}. \tag{7}$$

Note that multiplying a set by a scalar means that the scalar is multiplied by each element in the set. The addition of two sets means that the corresponding elements are added. This is only for mathematical expression, and $\rho$ in Eqn. (7) can be calculated by two forward passes through all the layers in practice. Then we obtain the gradient estimate for the $i$-th layer as

$$\widehat{\nabla}\mathcal{L}(\boldsymbol{W}_i; \mathcal{B}) = \rho\tilde{\boldsymbol{Z}}_i = \rho\boldsymbol{U}_i\boldsymbol{Z}_i\boldsymbol{V}_i^\mathsf{T}. \tag{8}$$

In Sec. 5, we analyze the effectiveness of this new gradient estimation (8). Specifically, Theorem 1 proves the close distance between our gradient estimate (8) and the vanilla gradient computed by BP in FO methods, while Theorem 2 shows smaller variance and angle error of our gradient estimate in Eqn. (8) compared to the gradient estimate (2) in MeZO (Malladi et al., 2023). See more theoretical details in Sec. 5.

Then, one can use estimated gradient in (8) to replace the gradient in any FO optimizer such as SGD:

$$\boldsymbol{W}_i^{t+1} = \boldsymbol{W}_i^t - \eta^t\widehat{\nabla}\mathcal{L}(\boldsymbol{W}_i^t; \mathcal{B}^t) = \boldsymbol{W}_i^t - \eta^t\rho^t\boldsymbol{U}_i^t\boldsymbol{Z}_i^t\boldsymbol{V}_i^{t\mathsf{T}}. \tag{9}$$

Here we choose SGD as the default optimizer of SubZero. Theorem 3 in Sec. 5 guarantees the convergence of SubZero with SGD as basic optimizer and gives its convergence rate. The choice of FO optimizers is orthogonal to ZO optimization. However, some empirical work indicates that adaptive optimizers like Adam (Kingma & Ba, 2015) do not necessarily enhance convergence of ZO approaches during LLM fine-tuning (Zhang et al., 2024; Guo et al., 2024). Also, there are other ZO optimizers that utilize stochastic momentum (Jiang et al., 2024) and second-order information (Zhao et al., 2024b) to facilitate faster convergence. While SubZero can be applied with other FO and ZO optimizers, we leave a comprehensive evaluation of these approaches for future work.

We compare the memory overhead of SubZero with the existing random subspace method S-RGF (Nozawa et al., 2024) using identical experimental settings, including layer-wise perturbation and matching subspace dimension, with all methods utilizing the SGD optimizer. As shown in Table 1, S-RGF's memory usage is roughly four times greater than SGD and 8.8 times that of MeZO (Malladi et al., 2023), while our SubZero's memory usage is comparable to MeZO. See more experimental comparison on OPT-13B in Table 5 of Sec. 6.

Table 1: Comparison of memory cost between SubZero and representative optimizers in full-parameter tuning scheme with RoBERTa-large on SST-2. "Mem." represents the total GPU memory cost.

| Method | Mem. (GB) |
|---|---|
| SGD | 6.063 |
| MeZO (Malladi et al., 2023) | 2.683 |
| S-RGF (Nozawa et al., 2024) | 23.845 |
| SubZero | 2.690 |

**Lazy Low-rank Subspace Update.** According to Eqn. (9), at the $t$-th iteration, the gradient estimate of the parameter matrix in the $i$-th layer, $\widehat{\nabla}\mathcal{L}(\boldsymbol{W}_i^t; \mathcal{B}^t)$, lies within a subspace defined by the projection matrices $\boldsymbol{U}_i^t$ and $\boldsymbol{V}_i^t$. Specifically, $\boldsymbol{U}_i^t$ spans the column subspace, while $\boldsymbol{V}_i^t$ determines the row subspace, with both matrices generated iteratively, leading to extra computational overhead to LLM fine-tuning.

However, for LLM fine-tuning, enhancing the computational efficiency and the accuracy of gradient subspace approximation is crucial. An excessively short update interval for $\boldsymbol{U}_i$ and $\boldsymbol{V}_i$, such as generating them iteratively, can incur high computational costs and limit exploration of the gradient subspace they established. Conversely, a long interval may result in inaccuracies in subspace approximation and fail to capture the evolving nature of the gradient subspace. Inspired by Galore (Zhao et al., 2024a), we propose a lazy subspace update strategy that periodically regenerates the projection matrices $\boldsymbol{U}_i$ and $\boldsymbol{V}_i$. Specifically, these matrices are generated at the first iteration of every $T_0 > 1$ training iterations and remain unchanged for the subsequent $T_0 - 1$ iterations (see lines 4-7 in Algorithm 3). We utilize QR decomposition on two different random matrices for generating the column-orthogonal matrices $\boldsymbol{U}_i$ and $\boldsymbol{V}_i$, as summarized in Algorithm 1. This lazy subspace update strategy is both efficient and effective in all our experiments.

**Algorithm 1** GenerateProjMatrix$(m, n, r)$

---

**Input:** size of parameter matrix $m \times n$, rank $r$.

1: Generate random matrices $\boldsymbol{R}_1 \in \mathbb{R}^{m \times r}$ and $\boldsymbol{R}_2 \in \mathbb{R}^{n \times r}$ whose entries are sampled from $\mathcal{N}(0,1)$
2: $\boldsymbol{U}, \_ \leftarrow$ QR_Decomposition$(\boldsymbol{R}_1)$
3: $\boldsymbol{V}, \_ \leftarrow$ QR_Decomposition$(\boldsymbol{R}_2)$
4: **return** $\boldsymbol{U}, \boldsymbol{V}$

---

**Algorithm 2** PerturbParams$(\mathcal{W}, \mathcal{U}, \mathcal{V}, r, \varepsilon, s)$

---

**Input:** model parameter set $\mathcal{W}$, projection matrix sets $\mathcal{U}$ and $\mathcal{V}$, rank $r$, perturbation scale $\varepsilon$, seed $s$.

1: Reset random number generator with seed $s$
2: **for** $i = 1, 2, \ldots, l$ **do**
3:     Generate the perturbation matrix $\boldsymbol{Z}_i \in \mathbb{R}^{r \times r}$ whose entries are sampled from $\mathcal{N}(0,1)$
4:     $\boldsymbol{W}_i \leftarrow \boldsymbol{W}_i + \varepsilon \boldsymbol{U}_i \boldsymbol{Z}_i \boldsymbol{V}_i^\mathsf{T}$
5: **return** $\mathcal{W}$

---

**Algorithm 3** SubZero

---

**Input:** parameter matrix in the $i$-th layer $\boldsymbol{W}_i \in \mathbb{R}^{m_i \times n_i}, i = 1, 2, \ldots, l$, loss $\mathcal{L}$, step budget $T$, perturbation scale $\varepsilon$, learning rate schedule $\{\eta^t\}$, subspace change frequency $T_0$, rank $r$.

1: **for** $t = 0, 1, \ldots, T-1$ **do**
2:     Sample a minbatch $\mathcal{B}^t \subset \mathcal{D}$ and a random seed $s^t$
3:     **for** $i = 1, 2, \ldots, l$ **do**
4:         **if** $t \mod T_0 \equiv 0$ **then**
5:             $\boldsymbol{U}_i^t, \boldsymbol{V}_i^t \leftarrow$ GenerateProjMatrix$(m_i, n_i, r)$
6:         **else**
7:             $\boldsymbol{U}_i^t \leftarrow \boldsymbol{U}_i^{t-1}, \boldsymbol{V}_i^t \leftarrow \boldsymbol{V}_i^{t-1}$
8:     // Note that $\mathcal{W}^t = \{\boldsymbol{W}_i^t\}_{i=1}^l, \mathcal{U}^t = \{\boldsymbol{U}_i^t\}_{i=1}^l, \mathcal{V}^t = \{\boldsymbol{V}_i^t\}_{i=1}^l$
9:     $\mathcal{W}^t \leftarrow$ PerturbParams $(\mathcal{W}^t, \mathcal{U}^t, \mathcal{V}^t, r, \varepsilon, s^t), \ell_+^t \leftarrow \mathcal{L}(\mathcal{W}^t; \mathcal{B}^t)$
10:    $\mathcal{W}^t \leftarrow$ PerturbParams $(\mathcal{W}^t, \mathcal{U}^t, \mathcal{V}^t, r, -2\varepsilon, s^t), \ell_-^t \leftarrow \mathcal{L}(\mathcal{W}^t; \mathcal{B}^t)$
11:    $\mathcal{W}^t \leftarrow$ PerturbParams $(\mathcal{W}^t, \mathcal{U}^t, \mathcal{V}^t, r, \varepsilon, s^t)$
12:    $\rho^t \leftarrow \left(\ell_+^t - \ell_-^t\right)/(2\varepsilon)$
13:    Reset random number generator with seed $s^t$
14:    **for** $i = 1, 2, \ldots, l$ **do**
15:       Regenerate the perturbation matrix $\boldsymbol{Z}_i^t \in \mathbb{R}^{r \times r}$ whose entries are sampled from $\mathcal{N}(0,1)$
16:       $\boldsymbol{W}_i^{t+1} \leftarrow \boldsymbol{W}_i^t - \eta^t \rho^t \left(\boldsymbol{U}_i^t \boldsymbol{Z}_i^t \boldsymbol{V}_i^{t\mathsf{T}}\right)$
17: **return** $\mathcal{W}^{t+1}$

---

SubZero maintains just three small matrices per layer: a perturbation matrix $\boldsymbol{Z}_i \in \mathbb{R}^{r \times r}$, and two column-orthogonal matrices $\boldsymbol{U}_i \in \mathbb{R}^{m_i \times r}$ and $\boldsymbol{V}_i \in \mathbb{R}^{n_i \times r}$. This design enhances memory efficiency, as $r$ is generally much smaller than the size of the corresponding parameter matrix $\boldsymbol{W}_i \in \mathbb{R}^{m_i \times n_i}$ (i.e., $r \ll \min\{m_i, n_i\}$). Morover, we employ in-place operations and per-layer parameter updates to estimate gradients and update parameters in parallel (see Appendix A.4). Consequently, SubZero uses significantly less GPU memory than previous methods while achieving similar or better performance. For example, fine-tuning OPT-1.3B (Zhang et al., 2022) on SST-2 (Socher et al., 2013b) using SGD (without momentum) in full-parameter scheme as shown in Table 3, SubZero requires only 6.8GB GPU memory, compared to 11.5GB for SGD, yielding a $1.6\times$ improvement in memory efficiency, similar as illustrated in Fig. 1 (d).

Now we are ready to summarize the overall algorithm of SubZero in Algorithm 3. Each training iteration consists of three main steps. First, it obtains the projection matrices $\boldsymbol{U}_i^t$ and $\boldsymbol{V}_i^t$ using Algorithm 1 or directly adopts previous ones. Next, it computes the loss value difference $\rho$ with Eqn. (7) by applying Algorithm 2 to perturb all parameter matrices. Finally, SubZero updates all parameter matrices layer by layer, following Eqn. (9).

### 4.2 INTEGRATION INTO FINE-TUNING SCHEMES

We describe the integration of SubZero into full-parameter tuning (Aghajanyan et al., 2021) and three promient PEFT schemes: LoRA (Hu et al., 2022), prefix tuning (Li & Liang, 2021), and prompt tuning (Lester et al., 2021). Typically, SubZero can be easily incorporated into these fine-tuning schemes. However, it encounters a challenge with extremely non-square parameter matrices, which have far more rows than columns or vice versa. This issue is particularly prevalent in LoRA,

which employs two low-rank matrices $\boldsymbol{A}_i \in \mathbb{R}^{m_i \times k}$ and $\boldsymbol{B}_i \in \mathbb{R}^{k \times n_i}$ to approximate a full matrix $\boldsymbol{W}_i' \in \mathbb{R}^{m_i \times n_i}$, with $k \ll \min\{m_i, n_i\}$, e.g., $k = 8$ while $\min\{m_i, n_i\} = 2048$ used in (Zhang et al., 2024). Consequently, it is impossible to find a smaller rank $r \ll k$ to compute the gradient estimates of $\boldsymbol{A}_i$ and $\boldsymbol{B}_i$ using Eqn. (6), imposing a challenge when applying SubZero to this scenario.

To overcome this limitation, we propose a reshaping strategy that transforms the original non-square matrix into an approximate square matrix. For instance, we reshape $\boldsymbol{A}_i \in \mathbb{R}^{m_i \times k}$ into $\boldsymbol{A}_i' \in \mathbb{R}^{m_i' \times k'}$ such that $m_i k = m_i' k'$ and $m_i'$ is close to $k'$. This reshaping allows us to apply Eqn. (6) to find a low-rank perturbation with rank $r$ significantly smaller than $\min\{m_i', k'\}$, demonstrating the applicability of SubZero in the scenario. Table 8 in Sec. 6.4 shows the effectiveness of this reshaping strategy.

## 5 THEORETICAL ANALYSIS

In this section, we theoretically analyze why SubZero can reduce the variance of gradient estimates and accelerate convergence. Before the analysis, we first define some necessary notations:

$$\boldsymbol{P} = \mathrm{bdiag}(\boldsymbol{V}_1 \otimes \boldsymbol{U}_1, \cdots, \boldsymbol{V}_l \otimes \boldsymbol{U}_l), \ \boldsymbol{z} = [\mathrm{vec}(\boldsymbol{Z}_1)^\mathsf{T}, \ldots, \mathrm{vec}(\boldsymbol{Z}_l)^\mathsf{T}]^\mathsf{T}, \ \tilde{\boldsymbol{z}} = [\mathrm{vec}(\tilde{\boldsymbol{Z}}_1)^\mathsf{T}, \ldots, \mathrm{vec}(\tilde{\boldsymbol{Z}}_l)^\mathsf{T}]^\mathsf{T}.$$

Then we first state the main theoretical results on our gradient estimation in Eqn. (8).

**Theorem 1.** *For the gradient estimation in Eqn. (8), the following two properties hold.*
*a) By using gradient estimation in (8), our estimated gradient $\hat{g}_\varepsilon(\boldsymbol{x}, \boldsymbol{P}, \boldsymbol{z})$ is equivalent to*

$$\hat{g}_\varepsilon(\boldsymbol{x}, \boldsymbol{P}, \boldsymbol{z}) = \frac{f(\boldsymbol{x} + \varepsilon \boldsymbol{P} \boldsymbol{z}) - f(\boldsymbol{x} - \varepsilon \boldsymbol{P} \boldsymbol{z})}{2\varepsilon} \boldsymbol{P} \boldsymbol{z}, \tag{10}$$

*where $\boldsymbol{z} \sim \mathcal{N}(\boldsymbol{0}, \boldsymbol{I}_q)$, $\varepsilon > 0$, $\boldsymbol{P} \in \mathbb{R}^{d \times q}$ satisfies $\boldsymbol{P}^\mathsf{T} \boldsymbol{P} = \boldsymbol{I}_q$ with $d = \sum_{i=1}^l m_i n_i$ and $q = lr^2$.*
*b) Let $\boldsymbol{z} \sim \mathcal{N}(\boldsymbol{0}, \boldsymbol{I}_q)$, and $f \in C_{L_2}^{2,2}(\mathbb{R}^d)$. Then we have*

$$\Phi(\boldsymbol{x}) = \|\mathbb{E}_{\boldsymbol{z}}[\hat{g}_\varepsilon(\boldsymbol{x}, \boldsymbol{P}, \boldsymbol{z})] - \boldsymbol{P} \boldsymbol{P}^\mathsf{T} \nabla f(\boldsymbol{x})\|_2 \leq \frac{\varepsilon^2}{6} L_2 (q + 4)^2.$$

See its proof in Appendix A.5. Theorem 1 (a) provides the equivalent form (10) of our gradient estimation (8). By comparing this with the gradient estimation (5) in random subspace optimization (Nozawa et al., 2024; Roberts & Royer, 2023), we observe significant differences. First, our gradient estimation (10) accounts for the layer-wise structure of the network, requiring the projection matrix $\boldsymbol{P}$ to be block-diagonal, whereas in random subspace optimization, $\boldsymbol{P}$ is not. Additionally, our method introduces a layer-wise low-rank perturbation matrix, reflected by the block-diagonal structure of $\boldsymbol{P}$, with lazy updates to the column and row spaces defined by $\boldsymbol{U}_i$ and $\boldsymbol{V}_i$. In contrast, random subspace optimization simply requires $\boldsymbol{P}$ to be random. These distinctions highlight the key differences between our gradient estimation and existing methods in random subspace optimization.

Theorem 1 (b) guarantees that the distance $\Phi(\boldsymbol{x})$ between the expected gradient estimate and the BP gradient in the subspace spanned by $\boldsymbol{P}$ is small. Moreover, by setting $\varepsilon = \frac{1}{q+4}$, the distance $\Phi(\boldsymbol{x})$ is bounded by a constant $L_2/6$, independent of the parameter dimension $d$. This implies that the error in our gradient estimation does not scale with the extremely high parameter dimensions of LLMs, providing highly accurate gradient estimation—crucial for optimizing LLMs.

Next, we utilize a strictly convex quadratic loss to further analyze our gradient estimation in Eqn. (10). This choice is motivated by the fact that, after pretraining, the LLM parameters tend to converge toward a local minimum within a local basin, which can be well-approximated by a quadratic loss (Neyshabur et al., 2020).

**Theorem 2.** *Let $f(\boldsymbol{x}) = \boldsymbol{x}^\mathsf{T} \boldsymbol{H} \boldsymbol{x}$ and $\boldsymbol{z} \sim \mathcal{N}(\boldsymbol{0}, \boldsymbol{I}_q)$, where $\boldsymbol{H} \in \mathbb{R}^{d \times d}$ is positive definite. We have*

$$\mathbb{E}_{\boldsymbol{z}}[\hat{g}_\varepsilon(\boldsymbol{x}, \boldsymbol{P}, \boldsymbol{z})] = \boldsymbol{P} \boldsymbol{P}^\mathsf{T} \nabla f(\boldsymbol{x}), \tag{11}$$

$$\mathbb{E}_{\boldsymbol{z}}[\|\hat{g}_\varepsilon(\boldsymbol{x}, \boldsymbol{P}, \boldsymbol{z})\|^2] = (q + 2)\|\boldsymbol{P}^\mathsf{T} \nabla f(\boldsymbol{x})\|^2, \tag{12}$$

$$\mathbb{E}_{\boldsymbol{z}} \left[ \frac{\langle \nabla f(\boldsymbol{x}), \hat{g}_\varepsilon(\boldsymbol{x}, \boldsymbol{P}, \boldsymbol{z}) \rangle^2}{\|\boldsymbol{P}^\mathsf{T} \nabla f(\boldsymbol{x})\|^2 \|\hat{g}_\varepsilon(\boldsymbol{x}, \boldsymbol{P}, \boldsymbol{z})\|^2} \right] = \frac{1}{q}. \tag{13}$$

See its proof in Appendix A.5. Theorem 2 demonstrates several advantageous properties of our gradient estimation on the quadratic function. First, Eqn. (11) establishes the equivalence between

Table 2: Performance of fine-tuning OPT-13B on SuperGLUE with various experimental settings (with 1000 examples). AVG: average score of all tasks.

| Task type | classification | | | | | | | multiple choice | | generation | | |
| Task | SST-2 | RTE | CB | BoolQ | WSC | WIC | MultiRC | COPA | ReCoRD | SQuAD | DROP | AVG. |
|---|---|---|---|---|---|---|---|---|---|---|---|---|
| SGD(FT) | 94.9 | 82.3 | 85.7 | 78.4 | 65.3 | 65.8 | 74.2 | 90.0 | 82.4 | 88.0 | 35.5 | 76.6 |
| Zero-shot | 58.8 | 59.6 | 46.4 | 59.0 | 38.5 | 55.0 | 46.9 | 80.0 | 81.2 | 46.2 | 14.6 | 53.3 |
| ICL | 87.0 | 62.1 | 57.1 | 66.9 | 39.4 | 50.5 | 53.1 | 87.0 | 82.5 | 75.9 | 29.6 | 62.8 |
| LP | 93.4 | 68.6 | 67.9 | 59.3 | 63.5 | 60.2 | 63.5 | 55.0 | 27.1 | 3.7 | 11.1 | 52.1 |
| MeZO(FT) | 92.1 | 71.5 | 71.4 | 74.4 | 61.5 | 60.0 | 60.1 | 87.0 | 82.0 | 84.2 | 31.2 | 70.5 |
| S-MeZO(FT) | 92.3 | **76.9** | **75.0** | **76.5** | 61.1 | 58.2 | **63.3** | 87.0 | 71.2 | 77.9 | 31.9 | 70.1 |
| SubZero(FT) | **92.1** | 74.0 | 73.2 | 75.3 | **65.4** | **60.8** | 61.0 | **88.0** | **82.3** | **84.5** | **32.0** | **71.5** |
| MeZO(LoRA) | 92.2 | 74.4 | 69.6 | 75.2 | 64.4 | 59.7 | 58.2 | 87.0 | **82.0** | 82.9 | 31.0 | 70.6 |
| S-MeZO(LoRA) | 90.8 | 62.2 | **75.0** | 72.9 | 51.9 | 55.8 | 56.4 | 86.0 | 69.9 | 76.4 | **31.7** | 66.3 |
| SubZero(LoRA) | **93.8** | **75.5** | 71.4 | **76.1** | **65.4** | 60.3 | 60.3 | **89.0** | 81.9 | **83.7** | 31.3 | **71.7** |

the expected gradient estimation and the BP gradient within the subspace spanned by our projection matrix $\boldsymbol{P}$. Second, Eqn. (12) shows that, in this subspace, the variance of the gradient estimation scales linearly with the subspace dimension $q$. In contrast, the variance of gradient estimation (2) in MeZO depends linearly on the model's parameter dimension $d$, which is significantly larger than $q$. Finally, Eqn. (13) reveals that the expected cosine similarity between our estimated gradient and the BP gradient within the subspace depends only on the subspace dimension $q \ll d$, indicating that our gradient estimation provides a highly accurate parameter update direction.

Building upon the above results, we can prove the convergence of our SubZero.

**Theorem 3.** *Let* $\boldsymbol{x}^* = \arg\min_{\boldsymbol{x} \in \mathbb{R}^d} f(\boldsymbol{x})$, *where* $f \in C_{L_1}^{1,1}(\mathbb{R}^d)$ *and* $f$ *is convex. Suppose* $\mathcal{E}_k = (\boldsymbol{e}_0, \cdots, \boldsymbol{e}_k)$ *where* $\boldsymbol{e}_k \sim \mathcal{N}(0, \boldsymbol{I}_q)$, $\eta = \frac{1}{4(q+4)L_1}$, $\phi_0 = f(\boldsymbol{x}_0)$, $\phi_k = \mathbb{E}_{\mathcal{E}_{k-1}}[f(\boldsymbol{x}_k)]$, $k \geq 1$ *where* $\{\boldsymbol{x}_k\}_{k>0}$ *is the sequence generated by Algorithm 3. For a fixed* $\boldsymbol{P}$, *then after* $N = \mathcal{O}(\frac{q}{\epsilon})$ *training iterations, we have*

$$\frac{1}{N+1} \sum_{k=0}^{N} (\phi_k - f^*) \leq \epsilon.$$

See its proof in Appendix A.5. Theorem 3 guarantees the convergence of our SubZero when the projection matrix $\boldsymbol{P}$ is fixed. Note, here we follow the approach of Galore (Zhao et al., 2024a), and assume a fixed projection matrix for simplicity. This convergence result can also be extended to cases where the projection matrix is lazily updated. Since lazy updates involve keeping the projection fixed over each periodic interval, we can prove convergence within each such period.

# 6    EXPERIMENTS

In this section, we present comprehensive experiments to evaluate the effectiveness of SubZero. We conduct our experiments using medium-sized masked LLMs (RoBERTa-large (Liu et al., 2019)) and large-scale autoregressive LLMs (OPT-1.3B and 13B (Zhang et al., 2022), LLaMA2-7B (Touvron et al., 2023), and Mistral-7B (Jiang et al., 2023)). Our exploration covers full-parameter tuning (FT) (Aghajanyan et al., 2021) and three PEFT schemes: LoRA (Hu et al., 2022), prefix tuning (Li & Liang, 2021), and prompt tuning (Lester et al., 2021). For comparison, we include leading ZO methods, such as MeZO (Malladi et al., 2023) and S-MeZO (Liu et al., 2024), alongside inference-only memory-efficient baselines like zero-shot, in-context learning (ICL) (Brown et al., 2020), and linear probing (LP) (Kumar et al., 2022). We also use the FO optimizer SGD as a benchmark. Since appropriate prompts are critical for ZO optimization (Malladi et al., 2023; Zhang et al., 2024), all experiments incorporate prompt templates, which are detailed in Appendix A.1.

## 6.1    PERFORMANCE WITH DIFFERENT EXPERIMENTAL SETTINGS

Following the settings in MeZO (Malladi et al., 2023), we evaluated SubZero using OPT-13B on the SuperGLUE benchmark (Wang et al., 2019), which covers a diverse range of tasks, including classification, multiple-choice, and generation, as outlined in Table 2. For each task, we randomly

Table 3: Performance of fine-tuning LLaMA2-7B and Mistral-7B on CB, and OPT-1.3B on SST-2.

| | LLaMA2-7B | | | | Mistral-7B | | | | OPT-1.3B | | | |
|---|---|---|---|---|---|---|---|---|---|---|---|---|
| | FT | LoRA | Prefix | Prompt | FT | LoRA | Prefix | Prompt | FT | LoRA | Prefix | Prompt |
| SGD | 69.6 | 75.0 | 69.6 | 69.6 | 73.2 | 75.0 | 69.6 | 62.5 | 93.2 | 93.0 | 93.1 | 90.7 |
| MeZO | 64.3 | 73.2 | 69.6 | 60.7 | 62.5 | 69.6 | 58.3 | 57.1 | 92.3 | 92.8 | 91.6 | 85.9 |
| SubZero | **71.4** | **75.0** | **76.8** | **66.1** | **64.3** | **73.2** | **64.3** | **62.5** | **93.4** | 92.9 | **92.2** | **89.1** |

Table 4: Fine-tuning performance comparison between SubZero and MeZO on RoBERTa-large and OPT-13B with non-differentiable objectives.

| Model | RoBERTa-large | | | | OPT-13B |
|---|---|---|---|---|---|
| Task | SST-2 | SST-5 | SNLI | MNLI | SQuAD |
| Zero-shot | 79.0 | 35.5 | 50.2 | 48.8 | 46.2 |
| Cross entropy (Adam) | 93.9 | 55.9 | 88.7 | 83.8 | 84.2 |
| Cross entropy (MeZO) | **92.9** | 53.2 | 83.0 | 77.0 | 84.2 |
| Cross entropy (SubZero) | **92.9** | **54.0** | **84.7** | **77.1** | **84.5** |
| Accuracy/F1 (MeZO) | 92.4 | 46.5 | 81.9 | 73.9 | 80.2 |
| Accuracy/F1 (SubZero) | **92.7** | **47.1** | **83.0** | **74.8** | **81.1** |

sampled 1000 examples for training, 500 for validation, and 1000 for testing. The ZO methods were applied to both full-parameter tuning (FT) and LoRA fine-tuning schemes, running for 20K steps.

Table 2 presents the key findings, highlighting the best-performing ZO method in bold. The results show that ZO techniques significantly outperform baseline approaches like zero-shot, in-context learning, and linear probing, underscoring their ability to enhance a pre-trained model's performance on downstream tasks.

From Table 2, one can also observer that SubZero consistently surpasses MeZO across all tasks and fine-tuning methods. For instance, SubZero boosts MeZO's accuracy from 61.1% to 65.4% on the WSC task (+4.3%) under FT, and from 58.2% to 60.3% on MultiRC using LoRA (+2.1%). S-MeZO demonstrated competitive performance on several classification tasks. However, SubZero outperformed S-MeZO in 6 out of 11 tasks with FT and 9 out of 11 tasks with LoRA. Additionally, SubZero's average score across all tasks was higher than S-MeZO's, which displayed inconsistent performance due to its selective parameter masking based on pre-determined thresholds—an approach that lacked robustness in practice. Moreover, S-MeZO's performance in the LoRA scheme was particularly poor, highlighting the need for more adaptive sparse masking strategies.

We further extended our evaluation of SubZero using OPT-1.3B, LLaMA2-7B, and Mistral-7B in FT and three PEFT schemes: LoRA, prefix tuning, and prompt tuning. As shown in Table 3, SubZero outperformed MeZO across all models and fine-tuning schemes. Notably, while MeZO struggled in the prompt tuning scheme, SubZero excelled, achieving performance levels that closely matched those of the SGD optimizer.

## 6.2 PERFORMANCE WITH NON-DIFFERENTIABLE OBJECTIVES

Following MeZO (Malladi et al., 2023), we respectively apply SubZero to fine-tune RoBERTa-large and OPT-13B using two non-differentiable objectives: accuracy and F1. As a baseline, we also report results using the cross-entropy objective with Adam. As shown in Table 4, SubZero consistently outperforms MeZO across both non-differentiable objectives and the cross-entropy benchmark, demonstrating its effectiveness across varying optimization goals.

## 6.3 MEMORY USAGE AND WALL-CLOCK TIME ANALYSIS

Table 5 compares the memory consumption and wall-clock time of ZO methods (MeZO and SubZero), SGD, and inference-only approaches (zero-shot and in-context learning (ICL)) using OPT-13B. Since inference-only methods do not involve fine-tuning, they have zero wall-clock time and their memory usage reflects only the inference load. For fine-tuning, all methods were run for 20K steps. The ZO methods, including SubZero, achieved over a 1.8× reduction in memory usage compared to

SGD. Notably, SubZero's memory footprint closely aligns with MeZO's, while offering improved performance.

Although SubZero introduces additional computational overhead for generating projection matrices via QR decomposition, this extra time represents less than 5% of the total wall-clock time. It is important to note that due to differences in how steps are defined between ZO methods and SGD, direct wall-clock time comparisons between the two are not entirely meaningful.

Table 5: Memory usage (GB) and wall-clock time (minutes) of fine-tuning OPT-13B, with SGD's batch size being 8 for SQuAD and 16 for other tasks.

| Task | SST-2 | | WIC | | SQuAD | |
|---|---|---|---|---|---|---|
| Method | Mem. | Time | Mem. | Time | Mem. | Time |
| Zero-shot/ICL | 24.2 | 0 | 24.8 | 0 | 27.2 | 0 |
| SGD(FT) | 48.9 | 190.3 | 48.9 | 257.3 | 122.7 | 623.7 |
| MeZO(FT) | 26.1 | 324.9 | 26.6 | 370.5 | 37.4 | 670.2 |
| SubZero(FT) | 26.5 | 337.3 | 27.1 | 385.3 | 37.8 | 690.5 |
| MeZO(LoRA) | 26.1 | 123.9 | 26.6 | 171.6 | 37.4 | 476.7 |
| SubZero(LoRA) | 26.1 | 130.3 | 26.6 | 179.7 | 37.4 | 486.5 |

## 6.4 ABLATION STUDY

We conducted a thorough investigation of the effectiveness of our proposed techniques. Table 6 shows that using a column-orthogonal projection matrix significantly outperforms a Gaussian random projection matrix, primarily due to the low-rank structure of the perturbation matrices. This low-rank perturbation is key to improving the quality of gradient estimation.

Next, Table 7 explores the effects of subspace rank $r$ and update frequency $T_0$ in Algorithm 3. The results demonstrate that SubZero is robust to variations in the subspace rank. However, performance drops sharply when the update frequency is too low, as the optimization becomes constrained to a single subspace for too long, limiting its adaptability.

Finally, Table 8 underscores the critical role of the reshaping strategy for handling highly non-square perturbation matrices, essential for ensuring effective perturbations in different layers of the model. Together, these results highlight the improvements brought by our design choices, particularly in terms of projection and reshaping strategies, and their impact on SubZero's robustness and performance.

Table 6: Orthogonal or random projection matrix.

| Dataset | Ortho. | Accuracy |
|---|---|---|
| RTE | ✗ | 67.5 |
| | ✓ | **74.0** |
| WSC | ✗ | 59.6 |
| | ✓ | **65.1** |

Table 7: Subspace change frequency $T_0$ and rank $r$.

| $T_0 \setminus r$ | 32 | 64 | 128 |
|---|---|---|---|
| 500 | **72.6** | 70.0 | 72.2 |
| 1000 | 73.6 | 71.8 | **74.0** |
| 2000 | 72.2 | **73.3** | 72.2 |
| 20000 | 70.4 | **71.1** | 68.6 |

Table 8: Reshaping strategy for non-square matrices on SST-2 with OPT-1.3B in PEFT schemes.

| Method | LoRA | Prefix | Prompt |
|---|---|---|---|
| MeZO | 92.8 | 91.6 | 85.9 |
| SubZero(w/o) | 92.1 | 89.4 | 74.2 |
| SubZero(w/) | **92.9** | **92.2** | **89.1** |

## 7 CONCLUSION

We have demonstrated that SubZero effectively fine-tunes large LLMs across various tasks and schemes with a memory cost comparable to that of inference. Extra experiments indicate that SubZero can optimize non-differentiable objectives. Our theory explains how SubZero reduces the variance of gradient estimates and accelerates convergence.

**Limitation.** In addition to the SGD optimizer, we have yet to explore combining SubZero with other first-order optimizers, such as Adam. While SubZero is also compatible with other memory-efficient techniques like parameter quantization (Li et al., 2024), we have not thoroughly investigated the practical effects of these combinations. We will leave these explorations for future work.

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

# A APPENDIX

## A.1 PROMPT TEMPLATES

For autoregressive LLMs, we have three task types: classification, multiple-choice, and question answering. We adopt the prompt templates for various tasks in (Malladi et al., 2023), which are summarized in Table 9. For masked LLMs, we also adopt the prompt templates in (Malladi et al., 2023) and present them in Table 10.

Table 9: The prompt templates used in the OPT-1.3B, OPT-13B, LLama2-7B, and Mistral-7B experiments.

| Task | Type | Prompt |
|------|------|--------|
| SST-2 | cls. | <text> It was terrible/great |
| RTE | cls. | <premise> 
 Does this mean that "<hypothesis>" is true? Yes or No? 
 Yes or No |
| CB | cls. | Does this mean that "<hypothesis>" is true? Yes or No? 
 Yes/No/Maybe |
| BoolQ | cls. | <passage> <question>? 
 Yes/No |
| WSC | cls. | <text> 
 In the previous sentence, does the pronoun "<span2>" refer to <span1>? Yes or No? 
 Yes/No |
| WIC | cls. | Does the word "<word>" have the same meaning in these two sentences? Yes, No? 
 <sentence1> 
 <sentence2> 
 Yes/No |
| MultiRC | cls. | <paragraph> 
 Question: <question> 
 I found this answer "<answer". Is that correct? Yes or No? 
 Yes/No |
| COPA | mch. | <premise> so/because <candidate> |
| ReCoRD | mch. | <passage> 
 <query>.replace("@placeholder", <candidate>) |
| SQuAD | QA | Title: <title> 
 Context: <context> 
 Question: <question> 
 Answer: |
| DROP | QA | Passage: <context> 
 Question: <question> 
 Answer: |

Table 10: The prompt templates used in RoBERTa-large experiments. $C$ is the number of classification categories.

| Task | $C$ | Type | Prompt |
|------|-----|------|--------|
| SST-2 | 2 | sentiment cls. | <sentence1> It was great/terrible |
| SST-5 | 5 | sentiment cls. | <sentence1> It was great/good/okay/bad/terrible |
| MNLI | 3 | NLI | <sentence1> ? Yes/Maybe/No , <sentence2> |
| SNLI | 3 | NLI | <sentence1> ? Yes/Maybe/No , <sentence2> |

## A.2 DATASETS

Following (Malladi et al., 2023), we use SuperGLUE (Wang et al., 2019) for OPT experiments, including BoolQ (Clark et al., 2019), CB (de Marneffe et al., 2019), COPA (Roemmele et al., 2011), MultiRC (Khashabi et al., 2018), ReCoRD (Zhang et al., 2018), RTE (Dagan et al., 2005; Bar Haim

et al., 2006; Giampiccolo et al., 2007; Bentivogli et al., 2009), WiC (Pilehvar & Camacho-Collados, 2019), and WSC (Levesque et al., 2012). We also utilize SST-2 (Socher et al., 2013a) and two question answering (QA) datasets, SQuAD (Rajpurkar et al., 2016) and DROP (Dua et al., 2019).

For LLama2-7B and Mistral-7B, we use CB (de Marneffe et al., 2019) in the full-parameter tuning and three PEFT schemes. For OPT-1.3B, we utilize SST-2 (Socher et al., 2013a) in the full-parameter tuning and three PEFT schemes.

For RoBERTa-large, we consider classification datasets: SST-2 (Socher et al., 2013a), SST-5 (Socher et al., 2013a), MNLI (Williams et al., 2018), and SNLI (Bowman et al., 2015). Following Malladi et al. (2023), the test set has 1000 examples for fast iteration, while we have 512 examples per class for both training and validation.

### A.3 HYPERPARAMETERS

Using a larger batch size can consistently reduce the variance in ZO optimization, thus enhancing fine-tuning performance (Malladi et al., 2023; Gautam et al., 2024; Yang et al., 2024). However, this increase in batch size also raises the time for forward passes and significantly elevates memory usage. We focus on developing ZO methods that minimize variance and improve performance with small batch sizes, with a default setting of 16. In some SGD experiments, like on MultiRC and SQuAD, the batch size is reduced to 8 due to limited GPU resources.

Consistent with previous studies (Malladi et al., 2023; Zhang et al., 2024; Liu et al., 2024; Yang et al., 2024), we employ SGD without momentum by default to maintain memory efficiency. SGD utilizes linear learning scheduling, while all ZO methods apply a constant learning rate, with weight decay set to 0.

For RoBERTa, we run Adam for 1K steps and ZO methods for 100K steps. In the rest experiments, we run Adam for 5 epochs and SGD and ZO methods for 20K steps.

We follow previous work to set the hyperparameters in the PEFT schemes (Malladi et al., 2023; Zhang et al., 2024). For LoRA, the rank is set to 8 and $\alpha$ is set to 16. For prefix tuning, the length of prefix tokens is set to 5, and we initialize these tunable representations by randomly sampling tokens from the vocabulary and then passing them through the LLM to get their keys and values at different attention layers. For prompt tuning, the length of prompt virtual tokens is set to 10, and the prompt tokens are initialized with actual token values from the model's embedding.

We present the hyperparameter search grids in Tables 11 and 12 to assist with result reproduction. For OPT-1.3B, we utilize the same hyperparameter settings as in Table 12. For Roberta-large, we use a learning rate of {1e-6, 5e-6} and $\varepsilon$=1e-3 for MeZO and SubZero, with a batch size of 64. The rank for SubZero is set to {8, 16, 24}, and subspace change frequency is adjusted to {1000, 2000}.

### A.4 IMPLEMENTATION DETAILS

We use one A800 GPU with the PyTorch 2.1.0+CUDA 11.8 framework for ZO methods and, if needed, two A800 GPUs for SGD.

The gradient estimation in SubZero is applicable to parameter matrices, while LLMs mainly consist of dense layers. For other trainable parameters, such as biases and layer normalization parameters, we recommend using the gradient estimation in MeZO (Malladi et al., 2023), as these layers contain fewer parameters.

We introduce two useful strategies to implement our SubZero efficiently in memory.

**In-place Operation.** As indicated in Eqn. (7), directly computing the loss difference $\rho$ requires twice the memory of inference, as it must store both the parameter matrix set $\mathcal{W}$ and the perturbation matrix set $\tilde{\mathcal{Z}}$. To mitigate this, we draw inspiration from MeZO and utilize in-place operations. By employing the random seed trick, we store a random seed to compute $\rho$ (see lines 9-12 in Algorithm 3 and Algorithm 2) and regenerate the low-dimensional perturbation matrices $\boldsymbol{Z}_1, \boldsymbol{Z}_2, \cdots, \boldsymbol{Z}_l$ (see line 15 in Algorithm 3). Consequently, the memory cost for fine-tuning with SubZero is nearly equivalent to that of inference (see Table 1 and Table 5).

Table 11: The hyperparameter search grids for OPT-13B. For each task, we run 20K steps for ZO methods (MeZO, S-MeZO, and SubZero) and 5 epochs for SGD. We record the best model checkpoint based on the validation loss every 500 training steps.

| Experiment | Hyperparameters | Values |
|---|---|---|
| MeZO(FT) | batch size | 16 |
| | learning rate | {1e-7, 2e-7, 5e-7, 1e-6} |
| | $\varepsilon$ | 1e-3 |
| MeZO(LoRA) | batch size | 16 |
| | learning rate | {1.5e-5, 3e-5, 5e-5} |
| | $\varepsilon$ | 1e-3 |
| S-MeZO(FT) | batch size | 16 |
| | learning rate | {1e-6, 5e-6} |
| | $\varepsilon$ | 1e-3 |
| | sparse rate | 0.75 |
| S-MeZO(LoRA) | batch size | 16 |
| | learning rate | {5e-5, 1e-4, 1e-3} |
| | $\varepsilon$ | 1e-3 |
| | Sparse rate | 0.75 |
| SubZero(FT) | batch size | 16 |
| | learning rate | {1e-7, 2e-7, 5e-7, 1e-6} |
| | $\varepsilon$ | 1e-3 |
| | rank | {32, 64, 128, 256 } |
| | subspace change frequency | {500, 1000, 2000} |
| SubZero(LoRA) | batch size | 16 |
| | learning rate | {1.5e-5, 3e-5, 5e-5} |
| | $\varepsilon$ | 1e-3 |
| | rank | {4, 8, 16} |
| | subspace change frequency | {500, 1000, 2000} |
| SGD(FT) | batch size | 16 |
| | Learning rate | {1e-4, 1e-3, 5e-3} |

**Per-layer Weight Update.** FO optimizers update all model parameters after BP by storing the entire gradients in memory. In contrast, ZO optimizers like SubZero calculate gradient estimates by first determining the loss value difference from two forward passes, then calculating the gradient estimate for each layer using this difference along with the layer's perturbation. To reduce memory usage during training, we can implement the parameter update with `optimizer.step()` after calculating the gradient estimate for each layer.

SubZero significantly reduces GPU memory consumption with the two implementation strategies. It should note that we use the per-layer weight update strategy for MeZO in all experiments.

To simplify hyperparameter tuning, we employ a norm alignment trick, allowing SubZero to directly utilize hyperparameter settings, such as the learning rate, from MeZO (Malladi et al., 2023). For a random perturbation matrix $\boldsymbol{Z} \in \mathbb{R}^{m \times n}$, and its low-rank approximation is $\hat{\boldsymbol{Z}} = \boldsymbol{U} \boldsymbol{Z}' \boldsymbol{V}^{\mathsf{T}}$, where $\boldsymbol{U} \in \mathbb{R}^{m \times r}$, $\boldsymbol{V} \in \mathbb{R}^{n \times r}$, and $\boldsymbol{Z}' \in \mathbb{R}^{r \times r}$. If $\boldsymbol{Z}$ and $\boldsymbol{Z}'$ are Gaussian random matrices, and $\boldsymbol{U}$ and $\boldsymbol{V}$ are column-orthogonal matrices, then we have:

$$\mathbb{E}[\|\boldsymbol{Z}\|_F] = \sqrt{\frac{m \times n}{r^2}} \mathbb{E}\left[\|\hat{\boldsymbol{Z}}\|_F\right].$$

Define $\mu = \sqrt{\frac{m \times n}{r^2}}$. Let MeZO's learning rate be $\eta$ and perturbation scale be $\varepsilon$. There are two equivalent approaches to obtain the perturbation for SubZero. The first approach involves multiplying the random low-dimensional perturbation matrix by $\mu$, with SubZero adopting MeZO's hyperparameters

Table 12: The hyperparameter search grids for LLama2-7B and Mistral-7B. For each task, we run 20K steps for ZO methods (MeZO and SubZero) and 5 epochs for SGD. We record the best model checkpoint based on the validation loss every 500 training steps.

| Experiment | Hyperparameters | Values |
|---|---|---|
| MeZO(FT) | batch size | 16 |
| | learning rate | {1e-7, 5e-7, 1e-6} |
| | $\varepsilon$ | 1e-3 |
| MeZO(LoRA) | batch size | 16 |
| | learning rate | {1e-6, 5e-6, 1e-5, 3e-5} |
| | $\varepsilon$ | 1e-3 |
| MeZO(Prefix) | batch size | 16 |
| | learning rate | {1e-3, 5e-3, 1e-2} |
| | $\varepsilon$ | 1e-1 |
| MeZO(Prompt) | batch size | 16 |
| | learning rate | {1e-3, 5e-3, 1e-2} |
| | $\varepsilon$ | 1e-1 |
| SubZero(FT) | batch size | 16 |
| | learning rate | {1e-7, 5e-7, 1e-6} |
| | $\varepsilon$ | 1e-3 |
| | rank | {24, 48} |
| | subspace change frequency | 1000 |
| SubZero(LoRA) | batch size | 16 |
| | learning rate | {1e-6, 5e-6, 1e-5, 3e-5} |
| | $\varepsilon$ | 1e-3 |
| | rank | {4, 8} |
| | subspace change frequency | 1000 |
| SubZero(Prefix) | batch size | 16 |
| | learning rate | {1e-3, 5e-3, 1e-2} |
| | $\varepsilon$ | 1e-1 |
| | rank | {4, 8} |
| | subspace change frequency | 1000 |
| SubZero(Prompt) | batch size | 16 |
| | learning rate | {1e-3, 5e-3, 1e-2} |
| | $\varepsilon$ | 1e-1 |
| | rank | {16, 24} |
| | subspace change frequency | 1000 |
| SGD(FT) | batch size | 16 |
| | Learning rate | {1e-5, 1e-4, 1e-3, 5e-3} |

directly: $\eta' = \eta$ and $\varepsilon' = \varepsilon$. The second approach keeps the random low-dimensional perturbation matrix fixed and sets SubZero's learning rate and perturbation scale as follows:

$$\eta' = \eta\mu^2, \varepsilon' = \varepsilon\mu.$$

We argue that norm alignment is crucial for SubZero, as changing the rank $r$ affects the norm of the gradient estimate, complicating the fine-tuning of the associated learning rate.

S-MeZO (Liu et al., 2024), a new ZO method, aims to improve MeZO's performance and convergence speed. However, its source code and detailed layer-wise hyperparameter configurations have not been released. Yang et al. (2024) reproduce S-MeZO using a fixed sparsity ratio for each layer, selected based on the best overall result shown in Fig. 6 of their paper. So we perform S-MeZO with this non-official implementation code available at https://github.com/yifanycc/AdaZeta.

A.5  PROOFS

In practice, SubZero employs smaller and layer-specific low-rank perturbation matrices instead of a large model-scale projection matrix. However, it is more convenient to prove SubZero's properties using a model-scale projection. Fortunately, the following lemma shows that the low-rank perturbation matrix for each layer can be represented as a layer-scale projection matrix, which is column orthogonal.

**Lemma 1.** *Let* $\tilde{\boldsymbol{Z}} = \boldsymbol{U} \boldsymbol{Z} \boldsymbol{V}^\mathsf{T}$, *where* $\boldsymbol{U} \in \mathbb{R}^{m \times r}, \boldsymbol{Z} \in \mathbb{R}^{r \times r}, \boldsymbol{V} \in \mathbb{R}^{n \times r}$, *and* $\boldsymbol{U}^\mathsf{T} \boldsymbol{U} = \boldsymbol{V}^\mathsf{T} \boldsymbol{V} = \boldsymbol{I}_r$. *Then we have* $\mathrm{vec}(\tilde{\boldsymbol{Z}}) = \boldsymbol{P} \mathrm{vec}(\boldsymbol{Z})$ *and* $\boldsymbol{P}^\mathsf{T} \boldsymbol{P} = \boldsymbol{I}_{r^2}$, *where* $\boldsymbol{P} = \boldsymbol{V} \otimes \boldsymbol{U}$.

*Proof.* Since $\mathrm{vec}(\boldsymbol{U} \boldsymbol{Z} \boldsymbol{V}^\mathsf{T}) = (\boldsymbol{V} \otimes \boldsymbol{U}) \mathrm{vec}(\boldsymbol{Z})$, we only need to show $(\boldsymbol{V} \otimes \boldsymbol{U})^\mathsf{T} (\boldsymbol{V} \otimes \boldsymbol{U}) = \boldsymbol{I}_{r^2}$. In fact

$$(\boldsymbol{V} \otimes \boldsymbol{U})^\mathsf{T} (\boldsymbol{V} \otimes \boldsymbol{U}) = (\boldsymbol{V}^\mathsf{T} \otimes \boldsymbol{U}^\mathsf{T})(\boldsymbol{V} \otimes \boldsymbol{U}) = (\boldsymbol{V}^\mathsf{T} \boldsymbol{V}) \otimes (\boldsymbol{U}^\mathsf{T} \boldsymbol{U}) = \boldsymbol{I}_r \otimes \boldsymbol{I}_r = \boldsymbol{I}_{r^2}.$$

The proof is completed. $\qquad\square$

We can also demonstrate that the low-rank perturbation matrices across all layers can be represented as a model-scale projection matrix. We first give the following lemma.

**Lemma 2.** *Let a block diagonal matrix* $\boldsymbol{P} = \mathrm{bdiag}(\boldsymbol{P}_1, \boldsymbol{P}_2, \cdots, \boldsymbol{P}_l)$ *and* $\tilde{\boldsymbol{z}}_i = \boldsymbol{P}_i \boldsymbol{z}_i$, *where* $\boldsymbol{P}_i^\mathsf{T} \boldsymbol{P}_i = \boldsymbol{I}_{r^2}$ *and* $i = 1, 2, \ldots, l$. *Then we have* $\tilde{\boldsymbol{z}} = \boldsymbol{P} \boldsymbol{z}$, *where* $\tilde{\boldsymbol{z}} = [\tilde{\boldsymbol{z}}_1^\mathsf{T}, \ldots, \tilde{\boldsymbol{z}}_l^\mathsf{T}]^\mathsf{T}$, $\boldsymbol{z} = [\boldsymbol{z}_1^\mathsf{T}, \ldots, \boldsymbol{z}_l^\mathsf{T}]^\mathsf{T}$ *and* $\boldsymbol{P}^\mathsf{T} \boldsymbol{P} = \boldsymbol{I}_{lr^2}$.

*Proof.* It is easy to check that $\tilde{\boldsymbol{z}} = \boldsymbol{P} \boldsymbol{z}$. Besides, we have

$$\boldsymbol{P}^\mathsf{T} \boldsymbol{P} = \mathrm{bdiag}(\boldsymbol{P}_1^\mathsf{T}, \ldots, \boldsymbol{P}_l^\mathsf{T}) \mathrm{bdiag}(\boldsymbol{P}_1, \ldots, \boldsymbol{P}_l) = \mathrm{bdiag}(\boldsymbol{P}_1^\mathsf{T} \boldsymbol{P}_1, \ldots, \boldsymbol{P}_l^\mathsf{T} \boldsymbol{P}_l) = \boldsymbol{I}_{lr^2}.$$

The proof is completed. $\qquad\square$

We may define $\boldsymbol{P} = \mathrm{bdiag}(\boldsymbol{V}_1 \otimes \boldsymbol{U}_1, \boldsymbol{V}_2 \otimes \boldsymbol{U}_2, \cdots, \boldsymbol{V}_l \otimes \boldsymbol{U}_l)$ that satisfies $\boldsymbol{P}^\mathsf{T} \boldsymbol{P} = \boldsymbol{I}$, $\boldsymbol{z} = [\mathrm{vec}(\boldsymbol{Z}_1)^\mathsf{T}, \mathrm{vec}(\boldsymbol{Z}_2)^\mathsf{T}, \ldots, \mathrm{vec}(\boldsymbol{Z}_l)^\mathsf{T}]^\mathsf{T}$, and $\tilde{\boldsymbol{z}} = [\mathrm{vec}(\tilde{\boldsymbol{Z}}_1)^\mathsf{T}, \mathrm{vec}(\tilde{\boldsymbol{Z}}_2)^\mathsf{T}, \ldots, \mathrm{vec}(\tilde{\boldsymbol{Z}}_l)^\mathsf{T}]^\mathsf{T}$. Then according to Lemma 2, the perturbation vector of SubZero is $\tilde{\boldsymbol{z}} = \boldsymbol{P} \boldsymbol{z}$, which is similar as existing random subspace methods in Eqn. (4), but with SubZero's projection matrix being block diagonal and column orthogonal.

To prove Theorem 1 and Theorem 2, we first introduce some definitions and lemmas about Gaussian distribution.

**Defination 1.** *We say* $\boldsymbol{z}$ *is a standard* $n$-*dimensional Gaussian vector (denote by* $\boldsymbol{z} \sim \mathcal{N}(\boldsymbol{0}, \boldsymbol{I}_n)$*), if its probability density function* $p(\boldsymbol{z}) = \frac{1}{\kappa} e^{-\frac{1}{2} \|\boldsymbol{z}\|^2}$, *where* $\kappa > 0$ *satisfies* $\int_{\mathbb{R}^n} \frac{1}{\kappa} e^{-\frac{1}{2} \|\boldsymbol{z}\|^2} d\boldsymbol{z} = 1$.

**Defination 2.** *Let* $\boldsymbol{z} \sim \mathcal{N}(\boldsymbol{0}, \boldsymbol{I}_n)$. *We say* $x$ *is a chi-square random variable with degrees of freedom* $n$ *(denote by* $x \sim \chi^2(n)$*), if* $x = \|\boldsymbol{z}\|^2$.

**Lemma 3.** *Let* $\boldsymbol{z} \sim \mathcal{N}(\boldsymbol{0}, \boldsymbol{I}_n)$. *For any orthogonal* $(n \times n)$-*matrix* $\boldsymbol{Q}$ *and continuous function* $f$, *we have* $\mathbb{E}_{\boldsymbol{z}}[f(\boldsymbol{z})] = \mathbb{E}_{\boldsymbol{z}}[f(\boldsymbol{Q}\boldsymbol{z})]$.

**Lemma 4.** *If* $x \sim \chi^2(n)$, *then we have*

$$\mathbb{E}_x[x] = n, \quad \mathrm{Var}_x[x] = 2n.$$

**Lemma 5.** *(Nesterov & Spokoiny, 2017) Let* $f \in C_{L_2}^{2,2}(\mathbb{R}^n)$. *Then for all* $\boldsymbol{x}, \boldsymbol{y} \in \mathbb{R}^n$, *we have*

$$\left| f(\boldsymbol{y}) - f(\boldsymbol{x}) - \langle \nabla f(\boldsymbol{x}), \boldsymbol{y} - \boldsymbol{x} \rangle - \frac{1}{2} \langle \nabla^2 f(\boldsymbol{x})(\boldsymbol{y} - \boldsymbol{x}), \boldsymbol{y} - \boldsymbol{x} \rangle \right| \le \frac{L_2}{6} \|\boldsymbol{y} - \boldsymbol{x}\|^3.$$

**Lemma 6.** *(Nesterov & Spokoiny, 2017) Let* $\boldsymbol{z} \sim \mathcal{N}(\boldsymbol{0}, \boldsymbol{I}_n)$. *For* $0 \le t \le 2$, *we have*

$$\mathbb{E}_{\boldsymbol{z}}[\|\boldsymbol{z}\|^t] \le n^{t/2}.$$

*For* $t \ge 2$, *we have*

$$n^{t/2} \le \mathbb{E}_{\boldsymbol{z}}[\|\boldsymbol{z}\|^t] \le (n + t)^{t/2}.$$

**Lemma 7.** *Let $\boldsymbol{z} \sim \mathcal{N}(\boldsymbol{0}, \boldsymbol{I}_n)$. For all $\boldsymbol{y} \in \mathbb{R}^n$, we have*

$$\mathbb{E}_{\boldsymbol{z}}[\|\langle \boldsymbol{y}, \boldsymbol{z} \rangle \boldsymbol{z}\|^2] = (n+2)\|\boldsymbol{y}\|^2.$$

*Proof.* Note that for any orthogonal $(n \times n)$-matrix $\boldsymbol{Q}$, we have

$$\|\langle \boldsymbol{y}, \boldsymbol{Q}\boldsymbol{z} \rangle \boldsymbol{Q}\boldsymbol{z}\|^2 = \|\langle \boldsymbol{Q}^\mathsf{T}\boldsymbol{y}, \boldsymbol{z} \rangle \boldsymbol{z}\|^2, \quad \|\boldsymbol{Q}^\mathsf{T}\boldsymbol{y}\| = \|\boldsymbol{y}\|.$$

In accordance with Lemma 3, we can set $\boldsymbol{y} = [1, 0, \ldots, 0]^\mathsf{T}$, and only need to prove $\mathbb{E}_{\boldsymbol{z}}[\|\langle \boldsymbol{y}, \boldsymbol{z} \rangle \boldsymbol{z}\|^2] = n + 2$. Equipped with Lemma 4, we get

$$\mathbb{E}_{\boldsymbol{z}}[\|\langle \boldsymbol{y}, \boldsymbol{z} \rangle \boldsymbol{z}\|^2] = \mathbb{E}_{\boldsymbol{z}}\left[\sum_{i=1}^n \boldsymbol{z}_1^2 \boldsymbol{z}_i^2\right] = \sum_{i=1}^n \mathbb{E}_{\boldsymbol{z}}[\boldsymbol{z}_1^2 \boldsymbol{z}_i^2] = \mathbb{E}_{\boldsymbol{z}_1}[\boldsymbol{z}_1^4] + \mathbb{E}_{\boldsymbol{z}_1}[\boldsymbol{z}_1^2] \sum_{i=2}^n \mathbb{E}_{\boldsymbol{z}}[\boldsymbol{z}_i^2] = n + 2.$$

The proof is completed. $\qquad\square$

**Theorem 1.** *For the gradient estimation in Eqn. (8), the following two properties hold.*
*a) By using gradient estimation in (8), our estimated gradient $\hat{g}_\varepsilon(\boldsymbol{x}, \boldsymbol{P}, \boldsymbol{z})$ is equivalent to*

$$\hat{g}_\varepsilon(\boldsymbol{x}, \boldsymbol{P}, \boldsymbol{z}) = \frac{f(\boldsymbol{x} + \varepsilon \boldsymbol{P}\boldsymbol{z}) - f(\boldsymbol{x} - \varepsilon \boldsymbol{P}\boldsymbol{z})}{2\varepsilon}\boldsymbol{P}\boldsymbol{z}, \tag{10}$$

*where $\boldsymbol{z} \sim \mathcal{N}(\boldsymbol{0}, \boldsymbol{I}_q)$, $\varepsilon > 0$, $\boldsymbol{P} \in \mathbb{R}^{d \times q}$ satisfies $\boldsymbol{P}^\mathsf{T}\boldsymbol{P} = \boldsymbol{I}_q$ with $d = \sum_{i=1}^l m_i n_i$ and $q = lr^2$.*
*b) Let $\boldsymbol{z} \sim \mathcal{N}(\boldsymbol{0}, \boldsymbol{I}_q)$, and $f \in C_{L_2}^{2,2}(\mathbb{R}^d)$. Then we have*

$$\Phi(\boldsymbol{x}) = \|\mathbb{E}_{\boldsymbol{z}}[\hat{g}_\varepsilon(\boldsymbol{x}, \boldsymbol{P}, \boldsymbol{z})] - \boldsymbol{P}\boldsymbol{P}^\mathsf{T}\nabla f(\boldsymbol{x})\|_2 \le \frac{\varepsilon^2}{6}L_2(q+4)^2.$$

*Proof.* **a)** Evidently, the conclusion is established based on Lemma 1 and Lemma 2.

**b)**

Let $a_{\boldsymbol{z}}(\tau) = f(\boldsymbol{x} + \tau\boldsymbol{z}) - f(\boldsymbol{x}) - \tau\langle \nabla f(\boldsymbol{x}), \boldsymbol{z} \rangle - \frac{\tau^2}{2}\langle \nabla^2 f(\boldsymbol{x})\boldsymbol{z}, \boldsymbol{z} \rangle$. Lemma 5 implies that

$$|a_{\boldsymbol{z}}(\pm\varepsilon)| \le \frac{\varepsilon^3}{6}L_2\|\boldsymbol{z}\|^3.$$

Note that

$$\mathbb{E}_{\boldsymbol{z}}[\hat{g}_\varepsilon(\boldsymbol{x}, \boldsymbol{P}, \boldsymbol{z})] - \boldsymbol{P}\boldsymbol{P}^\mathsf{T}\nabla f(\boldsymbol{x})$$
$$= \frac{\boldsymbol{P}}{2\kappa\varepsilon}\int_{\mathbb{R}^q}[f(\boldsymbol{x} + \varepsilon\boldsymbol{P}\boldsymbol{z}) - f(\boldsymbol{x} - \varepsilon\boldsymbol{P}\boldsymbol{z}) - 2\varepsilon\langle\nabla f(\boldsymbol{z}), \boldsymbol{P}\boldsymbol{z}\rangle]\boldsymbol{z}e^{-\frac{1}{2}\|\boldsymbol{z}\|^2}d\boldsymbol{z}.$$

Therefore, in accordance with Lemma 6, we have

$$\|\mathbb{E}_{\boldsymbol{z}}[\hat{g}_\varepsilon(\boldsymbol{x}, \boldsymbol{P}, \boldsymbol{z})] - \boldsymbol{P}\boldsymbol{P}^\mathsf{T}\nabla f(\boldsymbol{x})\|$$
$$\le \frac{1}{2\kappa\varepsilon}\int_{\mathbb{R}^q}|f(\boldsymbol{x} + \varepsilon\boldsymbol{P}\boldsymbol{z}) - f(\boldsymbol{x} - \varepsilon\boldsymbol{P}\boldsymbol{z}) - 2\varepsilon\langle\nabla f(\boldsymbol{z}), \boldsymbol{P}\boldsymbol{z}\rangle|\|\boldsymbol{z}\|e^{-\frac{1}{2}\|\boldsymbol{z}\|^2}d\boldsymbol{z}$$
$$= \frac{1}{2\kappa\varepsilon}\int_{\mathbb{R}^q}|a_{\boldsymbol{P}\boldsymbol{z}}(\varepsilon) - a_{\boldsymbol{P}\boldsymbol{z}}(-\varepsilon)|\|\boldsymbol{z}\|e^{-\frac{1}{2}\|\boldsymbol{z}\|^2}d\boldsymbol{z}$$
$$\le \frac{\varepsilon^2 L_2}{6\kappa}\int_{\mathbb{R}^q}\|\boldsymbol{z}\|^4 e^{-\frac{1}{2}\|\boldsymbol{z}\|^2}d\boldsymbol{z} \le \frac{\varepsilon^2}{6}L_2(q+4)^2.$$

The proof is completed. $\qquad\square$

**Theorem 2.** *Let $f(\boldsymbol{x}) = \boldsymbol{x}^\mathsf{T}\boldsymbol{H}\boldsymbol{x}$ and $\boldsymbol{z} \sim \mathcal{N}(\boldsymbol{0}, \boldsymbol{I}_q)$, where $\boldsymbol{H} \in \mathbb{R}^{d \times d}$ is positive definite. We have*

$$\mathbb{E}_{\boldsymbol{z}}[\hat{g}_\varepsilon(\boldsymbol{x}, \boldsymbol{P}, \boldsymbol{z})] = \boldsymbol{P}\boldsymbol{P}^\mathsf{T}\nabla f(\boldsymbol{x}), \tag{11}$$

$$\mathbb{E}_{\boldsymbol{z}}[\|\hat{g}_\varepsilon(\boldsymbol{x}, \boldsymbol{P}, \boldsymbol{z})\|^2] = (q+2)\|\boldsymbol{P}^\mathsf{T}\nabla f(\boldsymbol{x})\|^2, \tag{12}$$

$$\mathbb{E}_{\boldsymbol{z}}\left[\frac{\langle\nabla f(\boldsymbol{x}), \hat{g}_\varepsilon(\boldsymbol{x}, \boldsymbol{P}, \boldsymbol{z})\rangle^2}{\|\boldsymbol{P}^\mathsf{T}\nabla f(\boldsymbol{x})\|^2\|\hat{g}_\varepsilon(\boldsymbol{x}, \boldsymbol{P}, \boldsymbol{z})\|^2}\right] = \frac{1}{q}. \tag{13}$$

*Proof.* It is easy to check that $\hat{g}_\varepsilon(\boldsymbol{x}, \boldsymbol{P}, \boldsymbol{z}) = \boldsymbol{P}\langle \boldsymbol{P}^\mathsf{T}\nabla f(\boldsymbol{x}), \boldsymbol{z}\rangle\boldsymbol{z}$. Thus we have $\mathbb{E}_{\boldsymbol{z}}[\hat{g}_\varepsilon(\boldsymbol{x}, \boldsymbol{P}, \boldsymbol{z})] = \boldsymbol{P}\boldsymbol{P}^\mathsf{T}\nabla f(\boldsymbol{x})$. Combined with Lemma 7, we get $\mathbb{E}_{\boldsymbol{z}}[\|\hat{g}_\varepsilon(\boldsymbol{x}, \boldsymbol{P}, \boldsymbol{z})\|^2] = (q+2)\|\boldsymbol{P}^\mathsf{T}\nabla f(\boldsymbol{x})\|^2$. Note that for any orthogonal $(q \times q)$-matrix $\boldsymbol{Q}$, we have

$$\mathbb{E}_{\boldsymbol{z}}\left[\frac{\langle \nabla f(\boldsymbol{x}), \hat{g}_\varepsilon(\boldsymbol{x}, \boldsymbol{P}, \boldsymbol{z})\rangle^2}{\|\boldsymbol{P}^\mathsf{T}\nabla f(\boldsymbol{x})\|^2\|\hat{g}_\varepsilon(\boldsymbol{x}, \boldsymbol{P}, \boldsymbol{z})\|^2}\right] = \mathbb{E}_{\boldsymbol{z}}\left[\frac{\langle \boldsymbol{P}^\mathsf{T}\nabla f(\boldsymbol{x}), \boldsymbol{z}\rangle^2}{\|\boldsymbol{P}^\mathsf{T}\nabla f(\boldsymbol{x})\|^2\|\boldsymbol{z}\|^2}\right]$$

$$= \mathbb{E}_{\boldsymbol{z}}\left[\frac{\langle \boldsymbol{P}^\mathsf{T}\nabla f(\boldsymbol{x}), \boldsymbol{Q}\boldsymbol{z}\rangle^2}{\|\boldsymbol{P}^\mathsf{T}\nabla f(\boldsymbol{x})\|^2\|\boldsymbol{Q}\boldsymbol{z}\|^2}\right]$$

$$= \mathbb{E}_{\boldsymbol{z}}\left[\frac{\langle \boldsymbol{Q}^\mathsf{T}\boldsymbol{P}^\mathsf{T}\nabla f(\boldsymbol{x}), \boldsymbol{z}\rangle^2}{\|\boldsymbol{Q}^\mathsf{T}\boldsymbol{P}^\mathsf{T}\nabla f(\boldsymbol{x})\|^2\|\boldsymbol{z}\|^2}\right].$$

In accordance with Lemma 3, we can set $\boldsymbol{P}^\mathsf{T}\nabla f(\boldsymbol{x}) = [1, 0, \ldots, 0]^\mathsf{T}$. Thus we have

$$\mathbb{E}_{\boldsymbol{z}}\left[\frac{\langle \nabla f(\boldsymbol{x}), \hat{g}_\varepsilon(\boldsymbol{x}, \boldsymbol{P}, \boldsymbol{z})\rangle^2}{\|\boldsymbol{P}^\mathsf{T}\nabla f(\boldsymbol{x})\|^2\|\hat{g}_\varepsilon(\boldsymbol{x}, \boldsymbol{P}, \boldsymbol{z})\|^2}\right] = \mathbb{E}_{\boldsymbol{z}}\left[\frac{z_1^2}{\|\boldsymbol{z}\|^2}\right] = \frac{1}{q}.$$

The proof is completed. $\qquad\square$

**Lemma 8.** *Let $h(\boldsymbol{y}) = f(\boldsymbol{x} + \boldsymbol{P}\boldsymbol{y})$, where $f \in C_{L_1}^{1,1}(\mathbb{R}^d)$ and $f$ is convex, and $\boldsymbol{P}^\mathsf{T}\boldsymbol{P} = \boldsymbol{I}$, then we have $h \in C_{L_1}^{1,1}(\mathbb{R}^q)$ and $h$ is convex.*

*Proof.* Note that convexity is an affine-invariant property (Nesterov, 2018), if $f$ is convex, we can obtain that $h$ is also convex.

The following proves that if $f$ is first $L_1$-smooth, then $h$ is also first $L_1$-smooth. For any $\boldsymbol{y}_1 \in \mathbb{R}^q$ and $\boldsymbol{y}_2 \in \mathbb{R}^q$, we have

$$\|\nabla h(\boldsymbol{y}_1) - \nabla h(\boldsymbol{y}_2)\| = \left\|\boldsymbol{P}^\mathsf{T}\nabla(f(\boldsymbol{x} + \boldsymbol{P}\boldsymbol{y}_1) - \boldsymbol{P}^\mathsf{T}\nabla(f(\boldsymbol{x} + \boldsymbol{P}\boldsymbol{y}_2)\right\|$$

$$\leq \left\|\boldsymbol{P}^\mathsf{T}\right\|\|\nabla(f(\boldsymbol{x} + \boldsymbol{P}\boldsymbol{y}_1) - \nabla(f(\boldsymbol{x} + \boldsymbol{P}\boldsymbol{y}_2)\|$$

$$\leq L_1\|\boldsymbol{P}(\boldsymbol{y}_1 - \boldsymbol{y}_2)\|$$

$$= L_1\|\boldsymbol{y}_1 - \boldsymbol{y}_2\|$$

The proof is completed.

$\qquad\square$

**Theorem 3.** *Let $\boldsymbol{x}^* = \arg\min_{\boldsymbol{x}\in\mathbb{R}^d} f(\boldsymbol{x})$, where $f \in C_{L_1}^{1,1}(\mathbb{R}^d)$ and $f$ is convex. Suppose $\mathcal{E}_k = (\boldsymbol{e}_0, \cdots, \boldsymbol{e}_k)$ where $\boldsymbol{e}_k \sim \mathcal{N}(0, \boldsymbol{I}_q)$, $\eta = \frac{1}{4(q+4)L_1}$, $\phi_0 = f(\boldsymbol{x}_0)$, $\phi_k = \mathbb{E}_{\mathcal{E}_{k-1}}[f(\boldsymbol{x}_k)]$, $k \geq 1$ where $\{\boldsymbol{x}_k\}_{k>0}$ is the sequence generated by Algorithm 3. For a fixed $\boldsymbol{P}$, then after $N = \mathcal{O}(\frac{q}{\epsilon})$ training iterations, we have*

$$\frac{1}{N+1}\sum_{k=0}^{N}(\phi_k - f^*) \leq \epsilon.$$

*Proof.* In accordance with Lemma 8, we can transform the original problem $f \in C_{L_1}^{1,1}(\mathbb{R}^d)$ into $h \in C_{L_1}^{1,1}(\mathbb{R}^q)$ through affine transformation $h(\boldsymbol{y}) = f(\boldsymbol{x} + \boldsymbol{P}\boldsymbol{y})$. The subsequent convergence proof can directly refer to Theorem 8 in (Nesterov & Spokoiny, 2017).

The proof is completed. $\qquad\square$

