# OpenReview forum: "SubZero: Random Subspace Zeroth-Order Optimization for Memory-Efficient LLM Fine-Tuning"
_ICLR.cc/2025/Conference — ICLR 2025 Conference Withdrawn Submission_

### Official Review · Reviewer_q5sP · 2024-10-30

**Soundness:** 2
**Presentation:** 3
**Contribution:** 1
**Rating:** 3
**Confidence:** 4

**Summary:**

This paper proposes SubZero, a zeroth-order (ZO) optimization method tailored for memory-efficient LLM fine-tuning. The key innovation is using layer-wise low-rank perturbations to estimate gradients, combined with a lazy update strategy. The authors provide a theoretical analysis showing that their gradient estimates have lower variance than traditional ZO methods and prove convergence guarantees. Experimental results demonstrate improved performance over baselines like MeZO across LLM fine-tuning tasks while maintaining similar memory and runtime efficiency.

**Strengths:**

- The paper is clear and well written.
- The paper addresses an important problem, reducing computational requirements for fine-tuning/or training) LLMs.
- The paper contains a comprehensive theoretical analysis
- The paper considers two types of LLMS, autoregressive and masked.
- The method reduces variance in ZO optimization.

**Weaknesses:**

- The authors do not submit code to replicate the results.
- Another approach to reducing the variance is to increase the batch size. This can be done by gradient accumulation. However, the whole paper does not consider gradient accumulation as an approach to the variance or to reducing memory requirements. For example, in the experiments in Table 5, SGD with gradient accumulation could end up with the same memory/time requirements as MeZO.
- If the author's purpose is a new ZO method, it should be evaluated with existing ZO optimizers such as ZO-AdaMU (Jiang et al., 2024) or HiZOO (Zhao et al., 2024b). This comparison can not be something for future work.
- Same for Adam or **AdamW**, SGD lacks performance in LLM training/finetuning and AdamW is the default optimizer here (e.g. for LoRa). A comparison to AdamW is not a legitimate limitation and has to be considered in this work.
- The paper motivates the use of ZO or their SubZero approach with memory/runtime benefits in contrast to FO optimizers. A comparison of AdamW+LoRa, AdamW+FullParameter, and recent ZO methods **on the same compute budget** would be a required experimental setup (consider gradient accumulation to avoid different batch sizes).
- The paper states the importance of the batch size to ZO methods but does not analyze the impact of different batch sizes on the performance.
- The paper states that “a low-dimensional subspace may result in a reduced variance of the estimated gradient“ but lacks proof that a lower variance is beneficial for fine-tuning performance.
- In Table 2, one can not just take the AVG over scores in different regions. A better aggregation metric could be the average percentile performance improvement/reduction regarding a baseline (e.g. AdamW FT).

**Questions:**

- Is Table 1 based on batch size 1? It would be good to add AdamW and AdamW+LoRa to the table.
- Are the experiments on only one seed? What is the impact of the random seed on the proposed ZO method?
- In Table 3/4, what is the “performance” here?
- In Table 5, why is there no memory difference between FT and LoRa? Shouldn't LoRa reduce memory consumption?
- What is the impact of the batch size on the performance in SubZero?
- In Table 11, I appreciate the grid search for optimal hyperparameters but could you please provide the results of the watch experiment to show that the optimal solution is not in one end of the grid? Also, why is the search space for each method different?
- Is the grid search or the other experiments run with only one seed or multiple random seeds? If only one, have you tested the impact of random seed to a finetuning beforehand?

---

### Official Review · Reviewer_jKmz · 2024-11-04

**Soundness:** 2
**Presentation:** 2
**Contribution:** 1
**Rating:** 3
**Confidence:** 4

**Summary:**

The paper introduces SubZero, a method for fine-tuning large language models (LLMs) using random subspace zeroth-order (ZO) optimization. SubZero leverages random subspace perturbations and a low-rank approximation to estimate gradients without backpropagation, purportedly reducing memory usage. The authors claim that SubZero outperforms existing zeroth-order methods in terms of convergence and gradient variance while achieving comparable performance to first-order methods.

**Strengths:**

- **Memory efficiency**. The approach is valuable for scenarios where memory is a major constraint, as it does not require storing large gradients or optimizer states.
- **Theoretical result**. The mathematical backing provides a good foundation for the proposed method, with proof supporting claims on gradient variance and convergence properties.

**Weaknesses:**

- **Limited technical contributions**. Most of the improvements over MeZO and S-RGF of SubZero are taken from GaLore. For example, layer-wise and lazy low-rank update strategies. The integration into 4 different fine-tuning schemes is also studied in [1], leaving only the reshaping trick new to the best of my knowledge.
- **Narrow experimental scope.** The experiments are limited to a small set of benchmarks, which may not fully represent the method’s effectiveness across diverse LLMs or complex tasks. The focus on specific datasets does not showcase SubZero’s generalizability.
- **Application to Adam**. The limitations section of this paper notes that applying the method to the Adam optimizer is an area left for future investigation. However, fine-tuning LLMs with Adam typically results in better performance compared to SGD. For instance, Table 8 in [2] highlights a nearly 3% performance gap between fine-tuning with Adam and SGD. This raises a practical concern regarding the applicability of this method to real-world fine-tuning scenarios. Could the authors discuss the potential challenges or necessary modifications for applying SubZero with Adam?

**Questions:**

1. Figure 1a and Equation (13) indicate that the cosine similarity between the estimated gradient and the BP gradient is relatively low (nearly 0). The conclusion that the gradient estimation is effective is not convincing. Can the authors provide additional evidence or analysis to support their conclusion about the effectiveness of the gradient estimation?
2. While the reshaping technique improves performance, it is unclear how significantly it impacts the theoretical results. Could the authors provide an analysis or discussion on how the reshaping technique might impact the theoretical guarantees presented in the paper?
3. The SuperGLUE benchmark appears less challenging, as many tasks, such as classification and multiple-choice, are relatively straightforward. Fine-tuning and evaluating more complex benchmarks, such as those involving mathematical reasoning, would be more compelling. Examples include CommonSense170K [3] and MathInstruct [4].
4. Did the authors conduct experiments using multiple random seeds? It is unclear whether the improvements reported in some settings are statistically significant. Could the authors report the mean and standard deviation of the results across multiple random seeds?
5. Table 3 seems to showcase tasks where SubZero significantly outperforms MeZO as shown in Table 2. Additionally, CB is a relatively small dataset, with only 250 training samples and 55 validation samples, leading to a larger variance in fine-tuning results across different random seeds. Can the authors provide the results for more datasets?
6. Regarding Tables 2 and 4, which results are sourced from the MeZO paper, and which are original to this work? For instance, when comparing Table 4 of this paper with Table 3 in the MeZO paper, the Accuracy/F1 performance for MeZO appears to be lower in this study.
7. What are the experimental settings for Tables 6 and 7?
- Minor:
    - The subspace dimension is inconsistently denoted as both q and r. The authors should standardize the notation for clarity.

[1] Zhang, Yihua, et al. "Revisiting zeroth-order optimization for memory-efficient llm fine-tuning: A benchmark." *arXiv preprint arXiv:2402.11592* (2024).

[2] Xia, Mengzhou, et al. "Less: Selecting influential data for targeted instruction tuning." arXiv preprint arXiv:2402.04333 (2024).

[3] Hu, Zhiqiang, et al. "Llm-adapters: An adapter family for parameter-efficient fine-tuning of large language models." arXiv preprint arXiv:2304.01933 (2023).

[4] Yue, Xiang, et al. "Mammoth: Building math generalist models through hybrid instruction tuning." arXiv preprint arXiv:2309.05653 (2023).

---

### Official Review · Reviewer_ef3c · 2024-11-04

**Soundness:** 2
**Presentation:** 3
**Contribution:** 3
**Rating:** 5
**Confidence:** 3

**Summary:**

The paper introduces SubZero, a random subspace zeroth-order optimization method designed for memory-efficient fine-tuning of large language models (LLMs). Traditional backpropagation becomes impractical for such massive models due to high memory demands, and while zeroth-order (ZO) methods offer a memory-efficient alternative by estimating gradients using only forward passes, they suffer from high variance in high-dimensional settings typical of LLMs. SubZero addresses this issue by applying layer-specific low-rank perturbations, significantly reducing memory consumption and improving training performance. The authors theoretically prove that their gradient estimates closely approximate those from backpropagation and have lower variance than traditional ZO methods. They also introduce a simple yet effective pretraining strategy to implement SubZero effectively. Furthermore, they integrate SubZero into traditional and parameter-efficient fine-tuning techniques like LoRA, proposing specific adjustments to enhance this integration.

**Strengths:**

1. Clear and Well-Written: The paper is well-written, making complex concepts—including theoretical proofs—accessible and easy to understand.
2. Addresses a Critical Problem in Traditional LLM Fine-Tuning: It tackles the significant issue of high memory consumption during fine-tuning of large language models (LLMs). By maintaining only six matrices—a subset of the original full model—it substantially reduces memory requirements.
3. Effective Use of Zeroth-Order Optimization: The authors leverage existing zeroth-order (ZO) methods to approximate gradients efficiently. Their approach yields gradient estimates that are closer to true gradients and exhibit lower variance than traditional ZO methods.
4. Reproducibility Through Detailed Pseudocode: The inclusion of straightforward pseudocode and comprehensive methodological details ensures that the work is reproducible and easy to follow.
5. Comprehensive Ablation Studies: The paper provides thorough ablation studies on the components of the method. These experiments validate the effectiveness of each component and demonstrate their contributions to the overall performance.
6. Modularity and Integration with Existing Fine-Tuning techniques: It's commendable that the method is designed as a module that can be incorporated into both traditional fine-tuning and parameter-efficient fine-tuning methods like LoRA. The authors address issues arising from this integration by proposing practical techniques, investigating their validity and effectiveness, and ultimately delivering a robust and versatile method.
7. Strong Theoretical and Empirical Support: All claims are substantiated with theoretical proofs and empirical investigations.
8. Performance Improvements Over SoTA: The method shows performance boosts compared to existing state-of-the-art ZO methods, achieving faster convergence and better fine-tuning results across various language modeling tasks.
9. Evaluation on Diverse Downstream Tasks and Models: The authors use a variety of benchmarks and models to demonstrate the performance and ease of application of their method.

**Weaknesses:**

1. Lack of Comparison with Vanilla LoRA: The paper does not compare the proposed ZO-LoRA method directly with the standard LoRA approach, making it difficult to quantify the benefits of using ZO-LoRA over existing parameter-efficient fine-tuning methods. Including such a baseline would clarify the practical advantages of SubZero.
2. Missing Advanced LoRA Baselines: The evaluation does not consider advanced LoRA variants like AutoLoRA (Zhang et al., 2024). Including comparisons with such methods could strengthen the practical relevance of the paper.
3. Inconsistency in Reporting Results: In Table 2, for the SST-2 column under ZO-FT methods, the best-performing metric is not correctly highlighted; SubZero's metric is highlighted instead of the incumbent method's performance.
4. Influence of ReCoRD Task on Overall Performance: The ReCoRD task appears to disproportionately influence the average performance in the fine-tuning case. Excluding ReCoRD, the average scores for the methods become very similar (69.4, 70, and 70.4), making the differences negligible.
5. Unclear Computational Overheads and Budgets: In Figure 1c (training loss vs. wall-clock time), it is unclear whether the overhead associated with ZO methods is included. Additionally, the methods seem to have different computational budgets, complicating the comparison of convergence speeds and efficiency.
6. Need for Clarification on Variance Reduction: The paper emphasizes that SubZero reduces variance in gradient estimates and accelerates convergence. While it's generally understood that lower variance can lead to faster convergence, it's unclear why these are presented as two separate points. Clarifying this relationship would enhance understanding.

**Questions:**

1. Have you compared ZO-LoRA directly with vanilla LoRA, and can you provide the results?
2. Could you include comparisons with advanced LoRA variants like AutoLoRA to strengthen the practical side of your evaluation?
3. In Table 2, could you verify and correct the highlighting for the best metric in the SST-2 column under ZO-FT methods?
4. Could you explain the low performance of the ReCoRD task for S-MeZO?
5. In Figure 1c, does the wall-clock time include ZO methods' overhead, and why do the methods have different computational budgets?
6. Why are variance reduction and accelerated convergence presented as separate points when faster convergence is generally a result of lower variance in gradients?

---

### Official Review · Reviewer_rjAu · 2024-11-07

**Soundness:** 3
**Presentation:** 3
**Contribution:** 3
**Rating:** 6
**Confidence:** 4

**Summary:**

This paper presents SubZero, an innovative zeroth-order (ZO) optimization framework tailored for memory-efficient fine-tuning of large language models (LLMs). SubZero tackles the substantial memory requirements of conventional first-order optimizers, such as SGD, by employing a layer-wise low-rank perturbation technique for gradient estimation. This method not only reduces gradient variance but also achieves superior memory efficiency compared to other random projection methods.

**Strengths:**

- 1) The paper is well-written, presenting a range of experiments conducted across different datasets and model architectures.
- 2) It proposes a novel approach to avoid backpropagation, which accelerates the learning process and reduces gradient estimation variance without increasing memory consumption. This method offers significant benefits.
- 3) The theoretical convergence analysis is comprehensive and clearly explained. Notations are thoroughly specified, making the methodology easy to follow from the beginning, with detailed coverage of preliminaries.

**Weaknesses:**

- 1) The Related Work section requires a more comprehensive review. Numerous additional studies should be explored, particularly in the area of memory-efficient fine-tuning, where the discussed works also demand more in-depth coverage.
- 2) The memory comparison segment would be better placed outside the Methodology section.
- 3) Using \( T_0 \) to represent subspace update steps may cause confusion; considering alternative notations, like \( T \) or others, could improve reader comprehension.
- 4) The reshaping strategy needs to be tested across a broader range of scenarios to validate its effectiveness. The current explanations and experimental results do not sufficiently demonstrate its benefits.
- 5) Methods like GaLore and other recent approaches, which have outperformed LoRA on various tasks, should be included as baselines, for example, in Tables 2 and 3.
- 6) Given that SGD is not typically used as a state-of-the-art optimizer with LLMs today, the paper would benefit from comparisons with more advanced optimizers, such as Adam, in Tables 2 and 3, even if testing additional optimizers is left for future research.

**Questions:**

- 1) Why use the QR composition of two random matrices? Prior studies suggest that the gradient subspace aligns closely with the subspace of weight matrices. Could you investigate whether using the QR decomposition of weight matrices might enhance the performance of your proposed method?
- 2) A discussion on how the block-diagonal structure of your projection matrix contrasts with a random projection matrix could clarify how each impacts the different aspects of your proposed method.
- 3) Since a primary claim of this paper is to reduce the variance of other ZO methods, it would be helpful to include variance comparisons for other methods alongside the variance provided in Equation 12 for a more comprehensive and transparent analysis.
- 4) This paper assumes fixed projections in Theorem 3, similar to the approach taken in GaLore. However, unlike GaLore, which derives projections from data, your method uses random matrices to re-initialize projections. Additional details on how this "lazy" approach supports the convergence theory and why it works in random scenarios would strengthen the main text.

---

### Note · Authors · 2024-11-13

**Comment:**

We decided to withdraw the manuscript from ICLR 2025.

**Withdrawal Confirmation:**

I have read and agree with the venue's withdrawal policy on behalf of myself and my co-authors.